# SARS-CoV-2 infects adipose tissue in a fat depot- and viral lineage-dependent manner

Visceral adiposity is a risk factor for severe COVID-19, and a link between adipose tissue infection and disease progression has been proposed. Here we demonstrate that SARS-CoV-2 infects human adipose tissue and undergoes productive infection in fat cells. However, susceptibility to infection and the cellular response depends on the anatomical origin of the cells and the viral lineage. Visceral fat cells express more ACE2 and are more susceptible to SARS-CoV-2 infection than their subcutaneous counterparts. SARS-CoV-2 infection leads to inhibition of lipolysis in subcutaneous fat cells, while in visceral fat cells, it results in higher expression of pro-inflammatory cytokines. Viral load and cellular response are attenuated when visceral fat cells are infected with the SARS-CoV-2 gamma variant. A similar degree of cell death occurs 4-days after SARS-CoV-2 infection, regardless of the cell origin or viral lineage. Hence, SARS-CoV-2 infects human fat cells, replicating and altering cell function and viability in a depot- and viral lineage-dependent fashion.

The coronavirus disease 2019 (COVID-19) outbreak has become a significant public health emergency worldwide, yielding millions of victims. The COVID-19 etiologic agent is the novel enveloped RNA betacoronavirus named severe acute respiratory syndrome coronavirus 2 (SARS-CoV-2)[1]. Since its discovery in late 2019, SARS-CoV-2 has undergone several mutations and acquired distinct properties and capabilities, leading to the emergence of multiple variants, some of them characterized as variants of concern (VOCs) by the World Health Organization (WHO)[2,3].

Obesity increases the likelihood of the development of severe COVID-19, hence prolonging hospital stay and increasing death rates[4]. Besides the fact that individuals with obesity may display a compromised immune system, the higher abundance of adipose tissue in such a population has been pointed out as a major cause of severe COVID-19[4]. Since adipose tissue cells express *ACE2* and SARS-CoV-2 replication and its inflammatory insult are favored by the presence of lipid droplets[5], the hypothesis that adipose tissue may serve as a reservoir for storing and replicating the virus, as well as a site for cytokine amplification, has emerged as a potential explanation for the strong association between obesity and COVID-19 severity[6,7]. Recent studies have shown that SARS-CoV-2 can indeed infect adipose tissue cells, including adipocytes, thus favoring a local inflammatory response and resulting in changes in the lipid profile[8–12]. In turn, these changes are

thought to contribute to insulin resistance and hamper patients' recovery.

Such an "adipocentric" view is challenged because women usually have higher adiposity than men when paired by age[13]. However, conversely, the male population has been the most affected by the severe illness from COVID-19[14]. Indeed, fat distribution strikingly differs between sex as women tend to have more subcutaneous fat while men have higher visceral fat mass[13]. In line with this, the severe forms of COVID-19 are more specifically associated with high visceral fat mass than high overall adiposity or subcutaneous fat mass[15–18]. Furthermore, visceral adiposity is a stronger risk factor for cardiometabolic diseases than subcutaneous adiposity[19,20]. In this context, deciphering the contribution of different fat depots for SARS-CoV-2 infection and replication is imperative for understanding how adipose tissue contributes to COVID-19 pathophysiology.

Here we confirm that adipose tissue is a frequent extrapulmonary site where SARS-CoV-2 can be found in COVID-19 patients and provide evidence that visceral fat cells are more susceptible than subcutaneous fat cells to SARS-CoV-2 infection in vitro. We also show that SARS-CoV-2 triggers cellular responses that are partially overlapping between the fat depots but can also be depot-specific. Notably, SARS-CoV-2 results in higher levels of proinflammatory markers when it infects visceral fat cells when compared to subcutaneous fat cells. Finally, we observed

✉ e-mail: dmsouza@unicamp.br; mko@fmrp.usp.br; luizleiria@usp.br; morima@unicamp.br

that when visceral fat cells are infected with the SARS-CoV-2 gamma variant, viral load and cellular response are attenuated if compared to the effect of the SARS-CoV-2 ancient strain.

## Results

### SARS-CoV-2 infects human adipose tissue

To confirm that adipose tissue is infected in vivo, we studied *post-mortem* subcutaneous adipose tissue samples from the thoracic region obtained from individuals that died of COVID-19. Among 47 adipose tissue samples, SARS-CoV-2 RNA was detected in 23 (49%) (Supplementary Table 1). This is similar to what was found in other studies. One of which detected the viral RNA in visceral and/or subcutaneous adipose tissues of 15 out of 30 (50%) COVID-19 cases[10], while another found viral RNA in the subcutaneous adipose tissue of 13 out of 23 (56%) COVID-19 cases[12]. Interestingly, a study designed to investigate the organotropism of SARS-CoV-2 found that 46% of individuals deceased from COVID-19 had a systemic distribution of viral RNA, whereas the extrapulmonary tissue where SARS-CoV-2 RNA has been more frequently identified was the aorta (36%), followed by the small intestine (31%)[21]. Adipose tissue was not evaluated in this study. Adding to these findings, our results demonstrate that fat is a common extrapulmonary site where SARS-CoV-2 RNA can be detected in patients who died of COVID-19.

SARS-CoV-2 spike protein was found across the adipose tissue of subjects who had detectable viral genome in fat (Supplementary Fig. 1a–c), and adipocytes were among the cells infected, as demonstrated by SARS-CoV-2 spike protein detection around lipid droplets (Fig. 1a). DAPI and immunodetection of perilipin 1 and 2 were used as counterstaining, although perilipin 1 staining did not evidence lipid droplets as clear as perilipin 2 staining probably because of the autoptic nature of the samples and the presence of dead adipocytes and disrupted lipid droplets which seem to compromise perilipin 1 immunodetection in SARS-CoV-2-infected adipose tissue as previously demonstrated[11]. Viral load within adipose tissue (i.e., SARS-CoV-2 RNA normalized by host cell RNA) varied by several orders of magnitude among individuals, demonstrating considerable heterogeneity in the ability of fat to host SARS-CoV-2 (Fig. 1b). Adipose tissue viral load did not statistically associate with sex, body weight, BMI, or age (Fig. 1c–g), suggesting that the potential of SARS-CoV-2 to infect adipose tissue is independent of these COVID-19 risk factors. However, the lack of correlation may be due to the limited sample size. These results do not exclude the possibility that increased fat mass serves as a potentially more extensive reservoir for SARS-CoV-2, considering that our data reflects the relationship between SARS-CoV-2 RNA and host cell RNA. Thus, more cells in adipose tissue could represent a potentially larger reservoir for SARS-CoV-2. Moreover, these data do not take into account how cells from different fat depots enable SARS-CoV-2 infection and respond to it, given that our analyses up to this point were limited to the subcutaneous adipose tissue depot from the thoracic region. Indeed, visceral adiposity is a stronger predictor of COVID-19 severity if compared to overall adiposity or subcutaneous fat mass[15–18]. Finally, the detection of viral RNA or spike protein does not necessarily reflect the ability of the virus to persist and replicate in the tissue.

### Visceral fat cells are more susceptible to SARS-CoV-2 infection than subcutaneous fat cells

Considering these points, we turned to in vitro studies and tested the potential of SARS-CoV-2 to effectively enter and replicate in adipose cells derived from different anatomical regions. We compared fat depot differences in vitro using human primary stromal-vascular cells isolated from subcutaneous (abdominal) or visceral (omentum) adipose tissues from individuals undergoing abdominal surgery (i.e., cholecystectomy or bariatric surgery) and differentiated into adipocytes (Fig. 2a). That was based on the notion that differences between

fat depots are largely driven by the intrinsic characteristics of their progenitor cell populations and the ontogeny of these cells during adipogenesis[22–26]. Furthermore, by studying these cells in vitro in a reductionist manner and exposed to a high MOI (i.e., 1.0), we avoided potential extrinsic factors likely to influence SARS-CoV-2 infectious capacity, including virus availability. Moreover, we could minimize interindividual variability by using subcutaneous and visceral adipose tissue-derived cells isolated from the same individuals.

We differentiated these cells into adipocytes using a previously optimized protocol for primary human adipose tissue-derived stromal-vascular cells such as those we used in this study[27]. Extensive adipocyte differentiation was confirmed by Oil Red O staining and expression of adipocyte markers *FABP4* and *LEP*, and there were no differences when comparing differentiated visceral adipose tissue-derived cells (Vis AD) and differentiated subcutaneous adipose tissue-derived cells (Sub AD) (Supplementary Fig. 2a–c), thereby excluding a potential differentiation bias. We also confirmed the potential of these differentiated adipose tissue-derived cells to be infected by SARS-CoV-2, thus recapitulating what was found in vivo. Upon infection with an isolate of the original B SARS-CoV-2 lineage (obtained from the second case of COVID-19 in Brazil) [CoV-2(B)], the viral load increased over time in differentiated adipose tissue-derived cells (Supplementary Fig. 2d, e), demonstrating that SARS-CoV-2 can replicate in these cells. In addition, while trypsin incubation prior to harvesting drastically reduced viral load 24 h post-infection (hpi) in Vero cells—a bona fide model of SARS-CoV-2 infection and replication - it did not do so in adipose tissue-derived cells (Supplementary Fig. 2d), indicating that, at this time point, the quantity of viruses stuck outside adipose tissue-derived cells is negligible, whereas a large part of SARS-CoV-2 RNA detected in Vero cells is adhered extracellularly to the plasma membrane. Hence, we continued to pre-treat cells with trypsin before harvesting them to avoid external viral particles and properly compare adipose tissue-derived and Vero cells.

We confirmed the ability of CoV-2(B) to enter and replicate in both Vis AD and Sub AD cells using immunofluorescence detection of SARS-CoV-2 spike protein and double-stranded RNA (dsRNA) (Fig. 2b). Colocalization between spike and dsRNA was detected into the intracellular compartment and close to lipid droplets, indicating that SARS-CoV-2 can indeed infect and replicate in fully differentiated adipocytes (Fig. 2b). Remarkably, the viral load 24 hpi was 240-fold higher in Vis AD than in Sub AD cells (Fig. 2c). We performed plaque-forming assays using the conditioned medium of cells 24 hpi and found that Vis AD cells produced over ~770-fold more infectious particles than Sub AD cells (Fig. 2d). Cell viability was maintained until day 3 post-infection and then reduced to ~70% on days 4 and 5 in Vis AD and Sub AD cells (Supplementary Fig. 2f). These results demonstrate that although the degree of infection-induced cell death is similar when comparing cells from the two different adipose depots, adipose tissue cells originating from visceral fat are intrinsically more susceptible to SARS-CoV-2 infection than those from subcutaneous fat.

### Visceral fat cells express more ACE2 than subcutaneous fat cells

To decipher what could determine the potential of adipose tissue cells to be infected by SARS-CoV-2, we assessed the expression of genes encoding SARS-CoV-2 spike receptors and processing proteins in human deep-neck adipose tissue cells using publicly available single-nucleus RNA sequencing data[28]. While *ACE2*—the gene encoding the canonical SARS-CoV-2 spike receptor—was not detected in any of the cell types of deep-neck adipose tissue, the gene *NRP1*, which encodes another SARS-CoV-2 host cell infection mediator[29,30], was abundantly expressed in adipocytes, pre-adipocytes, macrophages, among others (Supplementary Fig. 3a, b). This raised the possibility that NRP1 mediates SARS-CoV-2 infection in adipose tissue. To test this, we incubated Vis AD or Sub AD cells

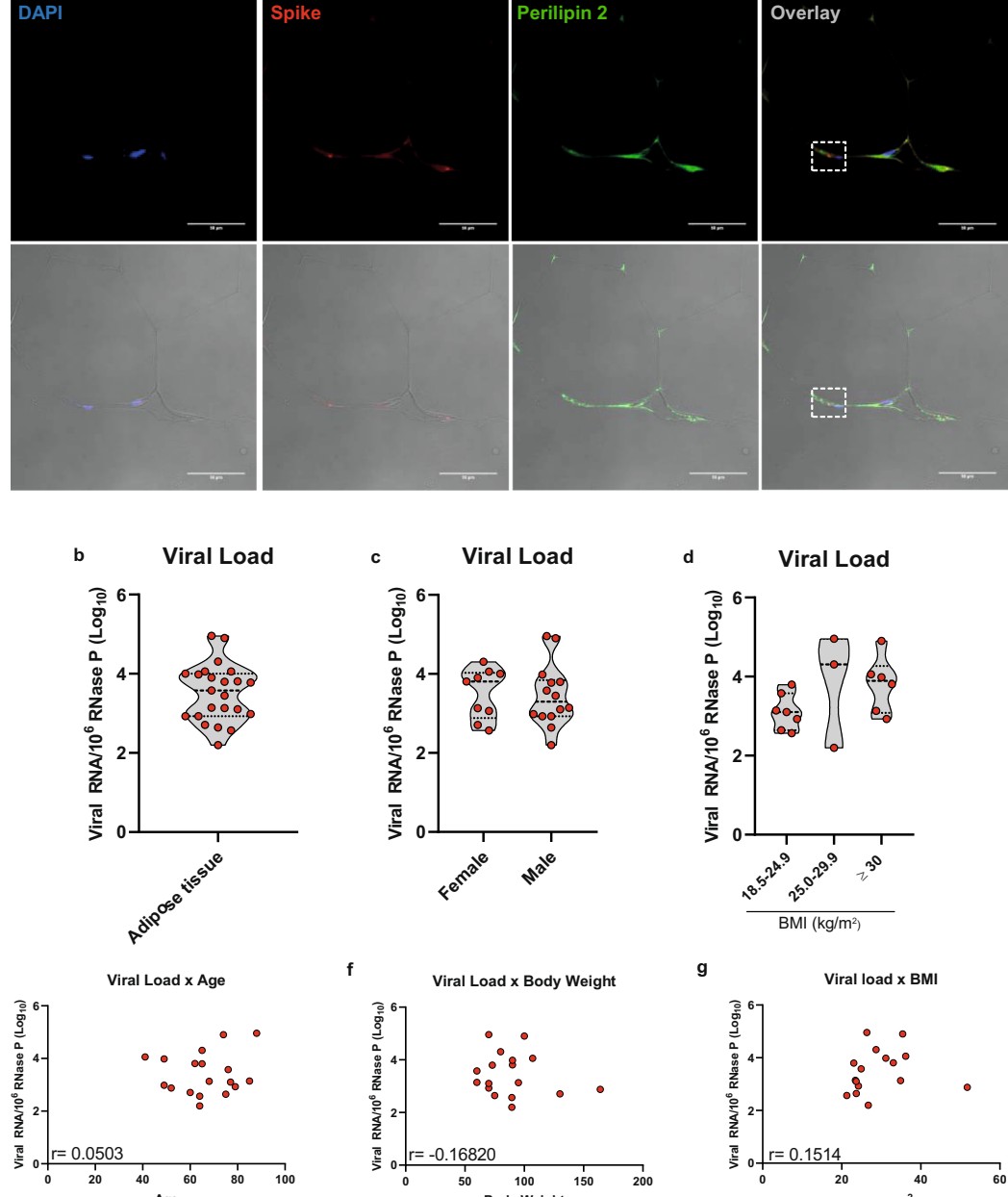

**Fig. 1 | SARS-CoV-2 infects human adipose tissue.** Thoracic subcutaneous adipose tissue samples of individuals who died of COVID-19. **a** Representative immuno-fluorescence images of 6–8 frames of two individuals. DAPI, blue. SARS-CoV-2 spike protein, red. Perilipin 2, green. The dashed box marks an area of an overlay with a concentration of spike labeling. The bottom panels are the overlay with the bright field. Scale bar = 50 μm. **b–d** Copy number of the SARS-CoV-2 genome normalized by $10^6$ copies of the endogenous RNase P gene as evaluated by RT-qPCR in the adipose tissue biopsies ($n = 23$) (**b**) and stratified by sex ($n = 23$) (**c**), and body mass index (BMI) ($n = 16$) (**d**). Violin plots show median, first and third quartiles of the means related to genomic copies, where each dot represents a biological sample.

Two-tailed Student's *t*-test and one-way ANOVA was applied in (**c**) and (**d**), respectively. No significant differences were found. **e–g** Correlation between copy number of the SARS-CoV-2 genome (normalized by $10^6$ copies of the endogenous RNase P gene) and age ($n = 18$) (**e**), body weight ($n = 17$) (**f**) and BMI ($n = 16$) (**g**). Only individuals that exhibited detectable levels of SARS-CoV-2 RNA in adipose tissue and had information about age, body weight, or BMI were used in the analysis (represented by dots in the correlation plot). Pearson correlation analysis was applied and no significant correlation was found. Source data are provided as a Source Data file and donor characteristics are shown in Supplementary Table 1 as averages. These experiments were repeated once with similar results.

with different types of NRP1 neutralizing antibodies or a selective NRP1 antagonist prior to and during SARS-CoV-2 exposure. No reduction in viral load at 24 hpi was observed (Supplementary Fig. 3c), demonstrating that NRP1 is not essential for infection.

This made us go back and look for ACE2 in adipose tissue cells using more sensitive methods. Indeed, Sub AD and Vis AD cells do express ACE2 as detected by immunofluorescence (Fig. 3a and Supplementary Fig. 4), western blotting (Fig. 3b), and RT-qPCR (Fig. 3c). Consistent with the increased susceptibility to SARS-CoV-2

infection, Vis AD cells expressed higher levels of ACE2 mRNA (Fig. 3c) and protein (Fig. 3b) when compared to Sub AD cells, indicating that the difference in the potential of these cells to be infected by SARS-CoV-2 is determined, at least in part, by the abundance of ACE2. In agreement with the dispensable role of NRP1, RT-qPCR data showed higher expression of *NRP1* in Sub AD compared to Vis AD cells (Fig. 3d), despite the higher viral load in the latter. There were no differences in *TMPRSS2* levels between the cells (Fig. 3e).

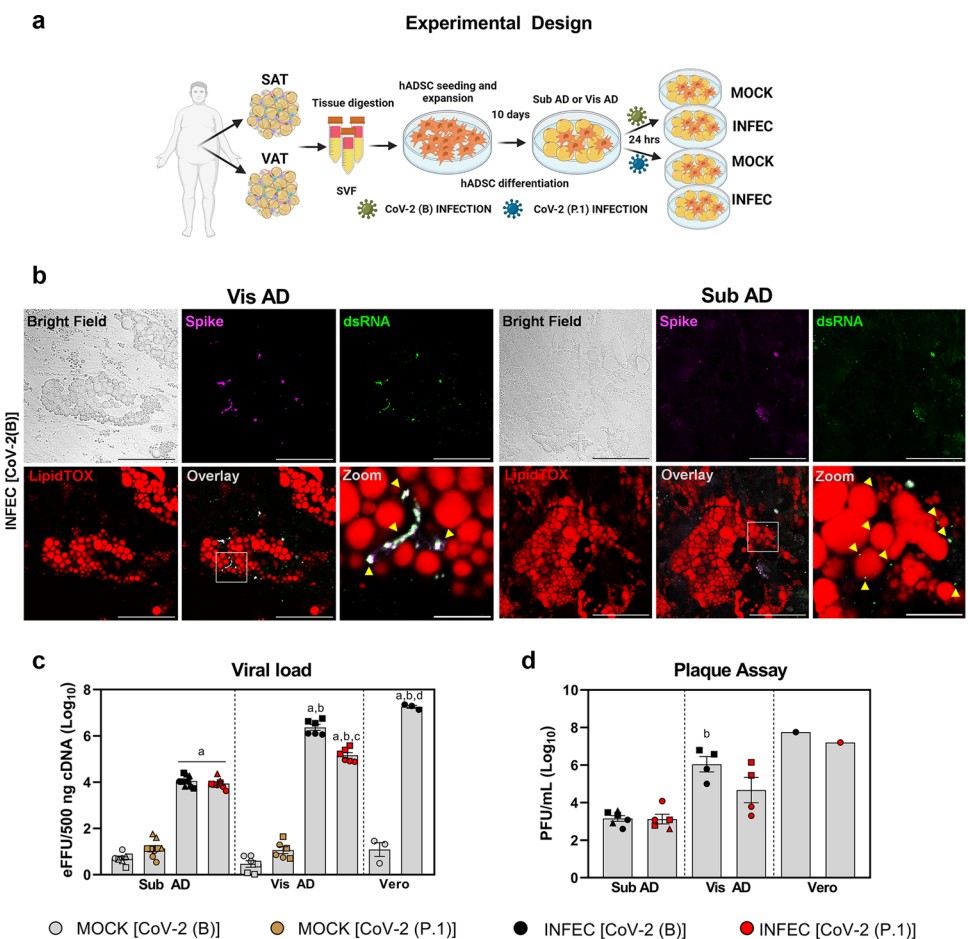

**Fig. 2 | The degree of adipose tissue cell infection depends on the anatomical origin of the cells and the SARS-CoV-2 lineage. a** Human adipose-derived stromal-vascular cells (hADSCs) were obtained from abdominal subcutaneous (SAT) and omental visceral (VAT) adipose tissues, differentiated in adipocytes (Sub AD and Vis AD, respectively), and infected (INFEC) or not (MOCK) with the SARS-CoV-2 ancient strain [CoV-2 (B)] or the P.1 variant [CoV-2 (P.1)]. Created with Biorender.com. **b** Cells were infected with MOI = 0.1 for 1 h and harvested 3 days after infection to allow better contrast and resolution of the replicating viruses. Representative immunofluorescence of six to ten images acquired from two to six independent pools of cells from two to three donors. SARS-CoV-2 spike protein, magenta. Double-stranded RNA (dsRNA), green. LipidTOX, red. Scale bar = 50 μm (zoom = 10 μm). **c, d** Cells were harvested 24 h after infection with MOI = 1 for 1 h to assess their susceptibility to viral infection under conditions where virus availability is not limiting. **c** Viral load determined by RT-qPCR. eFFU, equivalent to focus forming units. Three-way ANOVA was performed to determine effects of infection ($P < 0.0001$), viral lineage ($P > 0.05$), anatomical origin ($P < 0.0001$), and

interactions between anatomical origin × infection ($P < 0.0001$), anatomical origin × viral lineage ($P < 0.01$), infection × viral lineage ($P < 0.0001$), and infection, anatomical origin, and viral lineage ($P < 0.0001$). Significant differences between groups were determined by Sidak's multiple comparison test and depicted as: a, $P < 0.0001$ vs. MOCK; b, $P < 0.0001$ vs. Sub AD; c, $P < 0.0001$ vs. CoV-2(B); d, $P < 0.0001$ vs. Vis AD. **d** Plate forming units (PFU) in the supernatant. Two-way ANOVA was performed to determine the effects of anatomical origin ($P < 0.0001$), viral lineage ($P > 0.05$), and the interaction between anatomical origin and viral lineage ($P > 0.05$). Significant differences between groups were determined by Tukey's multiple comparison test and depicted as: b, $P < 0.01$ vs. Sub AD. Data were mean ± SEM of two to three independent pools of cells from two to three donors. Vero cells were used as reference. Different donors are distinguished by circles, squares, and triangles, while independent pools of cells from the same donor (or Vero) are marked by the same symbol. Source data are provided as a Source Data file. These experiments were repeated once with similar results.

## Lower viral load in visceral fat cells upon infection by the SARS-CoV-2 gamma variant P.1

Considering the evolution of SARS-CoV-2 and how genetic variants determine its infectiveness, we asked whether differentiated adipose tissue-derived cells were similarly susceptible to a variant of concern (VOC)−i.e., the gamma variant P.1 [CoV-2(P.1)]. This variant first appeared in Manaus, Brazil, and was responsible for most infections and deaths by COVID-19 in this country during 2021[3]. In Vis AD cells, the viral load of CoV-2(P.1) at 24 hpi was 17 times lower than that of CoV-2(B), but still 35-fold higher than in Sub AD cells (Fig. 2c). The number of infectious particles in the medium followed a similar pattern, although the differences were not statistically significant ($p = 0.0935$ when comparing Sub AD vs. Vis AD cells, Fig. 2d). There were no differences between CoV-2(B) and CoV-2(P.1) viral load in Sub AD cells (Fig. 2c, d). These results show that a VOC lineage of SARS-

CoV-2 can also infect differentiated adipose tissue-derived cells, although in a less efficient fashion depending on the depot. Regardless of the viral load at 24 hpi, CoV-2(P.1) led to a level of cell death in both Sub AD and Vis AD cells that was similar to the one induced by CoV-2(B) (Supplementary Fig. 2f).

## Distinct cellular responses in visceral and subcutaneous fat cells upon infection with different SARS-CoV-2 lineages

To understand the molecular changes in Vis AD and Sub AD cells in response to infection by different SARS-CoV-2 lineages, we conducted quantitative proteomics analyses. Infection of Vis AD and Sub AD cells with the CoV-2(B) lineage resulted in pronounced changes in the proteome at 24 hpi, with the majority of differentially expressed proteins being downregulated (Supplementary Figs. 5a, b, 6a, b and Supplementary Data 1). We identified 1415 proteins in Vis AD cells, of which

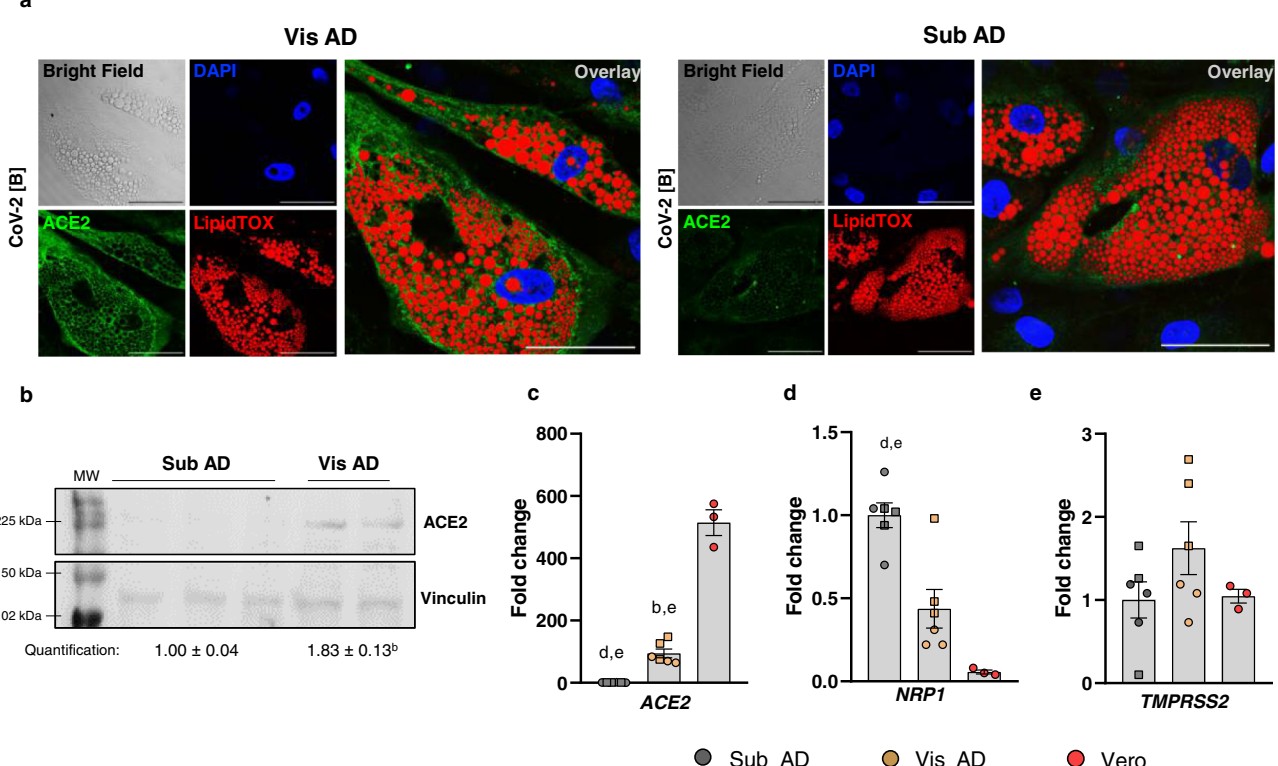

**Fig. 3 | Visceral adipose tissue cells express more ACE2 than subcutaneous adipose tissue cells.** Human adipose tissue-derived stromal-vascular cells from subcutaneous and visceral fat depots were differentiated into adipocytes (Sub AD and Vis AD, respectively). **a** Representative immunofluorescence of Vis AD and Sub AD cells infected with CoV-2(B) (MOI = 0.1 for 1 h and harvested 3 days after infection) evidencing ACE2 expression. DAPI, blue. ACE2, green. LipidTOX, red. Scale bar = 50 μm. Representative of eight to ten images acquired from two to five independent pools of cells from two to three donors. **b** Western blotting comparing the basal levels of ACE2 in uninfected Sub AD (*n* = 3 donors) and Vis AD (*n* = 2 donors) cells. Two-tailed Student's *t*-test: b, *P* < 0.01. MW, molecular weight. **c–e** RT-qPCR showing basal relative expression of genes related to SARS-CoV-2 entry in uninfected Sub AD and Vis AD cells. One-way ANOVA was applied in order to detect the effect of anatomical origin on **c** *ACE2* (*P* < 0.0001), **d** *NRP1* (*P* < 0.0001), and

**e** *TMPRSS2* (*P* > 0.05) expression. Tukey's multiple comparison test was applied to detect differences between the groups. Significant differences are depicted in letters: b, *P* < 0.01 vs. Sub AD; d, *P* < 0.01 vs. Vis AD; e, *P* < 0.0001 vs. Vero. Data represent the mean ± SEM of experiments performed using three independent pools of cells from two to three donors. Vero cells were used as a reference. Different donors are distinguished by circles, squares, and triangles, while independent pools of cells from the same donor (or Vero) are marked by the same symbol. Source data are provided as a Source Data file. The experiments were repeated once with similar results, with the exception of **c–e**, which were not repeated; although the calculated interassay variability was low, the controls worked as expected, and the biological and technical variability was accounted for in the analysis; moreover, the main findings were corroborated by **a**, **b** using other approaches.

---

901 were found to be significantly altered by CoV-2(B), and 1719 proteins in Sub AD cells, of which 774 were found to be significantly altered by CoV-2(B). There was a 76% overlap between the proteins identified in Vis AD vs. Sub AD, although when compared to the proteins altered by infection, the overlap decreased to 34% (i.e., 423 proteins). The degree of overlap between the Vis AD and Sub AD cellular responses was considerable, but there were also depot-specific responses (Fig. 4a, b), suggesting that SARS-CoV-2 may trigger different pathways in cells from different fat depots. CoV-2(P.1) infection also led to an overall downregulation of proteins in Sub AD cells (Supplementary Figs. 5c, 6c and Supplementary Data 1), but the opposite was observed in Vis AD cells (Supplementary Figs. 5d, 6d and Supplementary Data 1). We identified 1446 proteins in Vis AD cells, of which 836 were significantly altered by the CoV-2(P.1) variant, and 1521 proteins in Sub AD cells, of which 712 were significantly affected by CoV-2(P.1). There was 61% overlap between the proteins identified in Vis AD vs. Sub AD in this case, although the overlap between the significantly altered proteins was only 22% (i.e., 292 proteins). The degree of overlap with proteins altered by the CoV-2(B) infection was smaller than the lineage-specific responses (Fig. 4a). Even so, in terms of enriched ontology, infection of Sub AD cells by CoV-2(B) or CoV-2(P.1) resulted in more similar responses than that of Vis AD cells infected with CoV-2(B) or CoV-2(P.1) (Fig. 4b). These results demonstrate that

the responses of adipose tissue cells to SARS-CoV-2 infection are largely fat depot and viral lineage dependent.

## SARS-CoV-2 inhibits lipolysis in subcutaneous fat cells

The proteins altered in Sub AD and Vis AD cells in response to infection with the CoV-2(B) or CoV-2(P.1) lineages were commonly associated with the terms "Fatty acid metabolism" (Fig. 4b) and "Metabolic pathways" (Supplementary Fig. 7 and Supplementary Data 2). The terms "Pyruvate metabolism" and "Oxidative phosphorylation", on the other hand, were specifically enriched among the proteins altered by infection in Sub AD cells (Supplementary Fig. 7). Among the differentially expressed proteins involved in fatty acid metabolism, downregulation of hormone-sensitive lipase (HSL or LIPE) in infected Sub AD cells (Fig. 4c and Supplementary Data 1) caught our attention due to the fundamental role this enzyme plays in lipolysis[31]. Downregulation of HSL and its phosphorylated and active form was confirmed by western blotting in infected Sub AD cells (Fig. 4d). Another critical enzyme in lipolysis—adipose triglyceride lipase (ATGL)—was also downregulated by SARS-CoV-2 (Fig. 4d). Consistent with inhibition of lipolysis, glycerol production was decreased in Sub AD cells infected with SARS-CoV-2 (Fig. 4e). A previous study demonstrated that SARS-CoV-2 induces lipid droplet formation in monocytes, contributing to viral replication[5]. Moreover, pharmacological inhibition of lipolysis

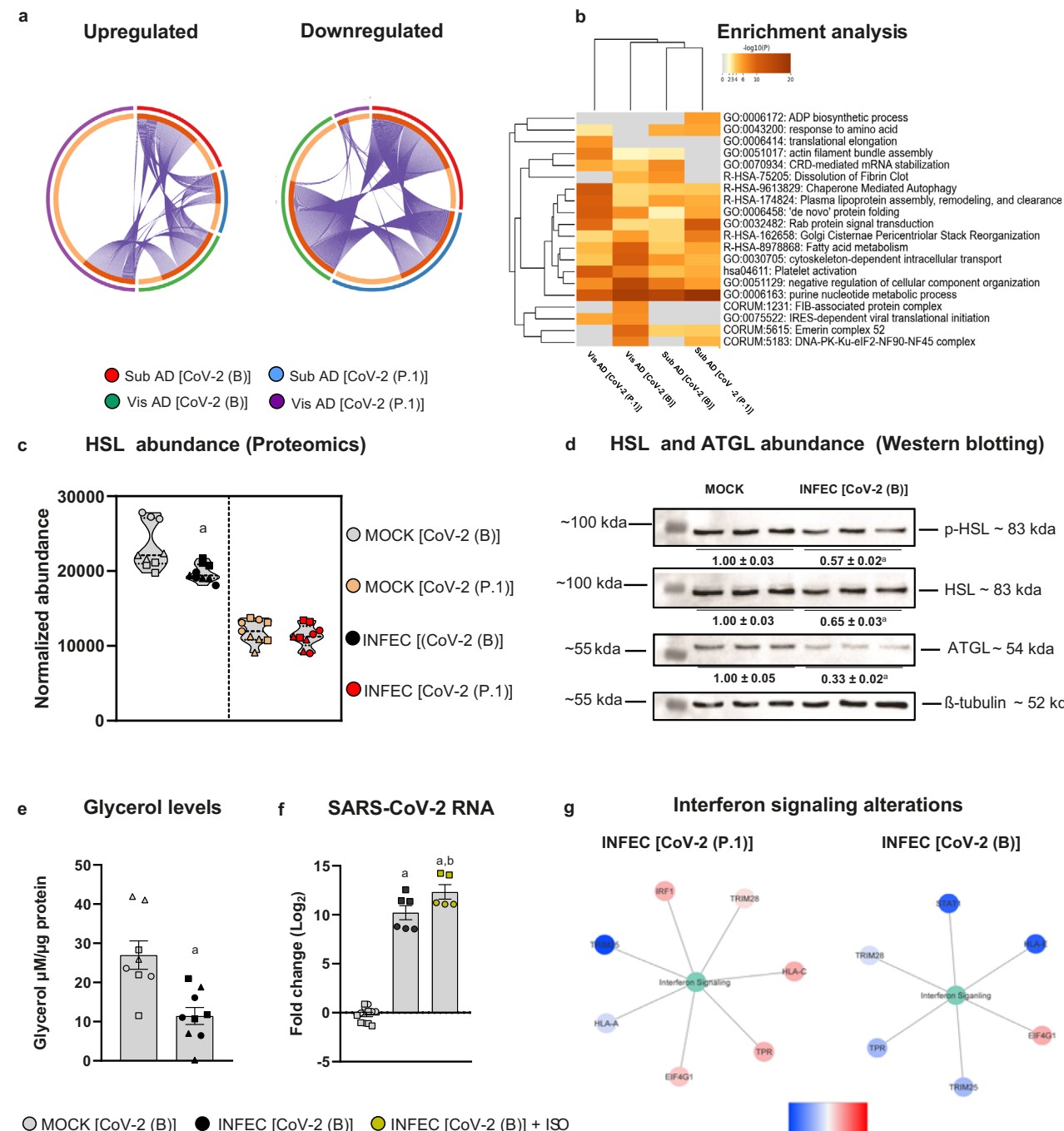

**Fig. 4 | SARS-CoV-2 infection of fat cells results in different cellular responses depending on the depot of origin and the viral lineage.** Stromal-vascular cells from human subcutaneous and visceral fat depots were differentiated into adipocytes (Sub AD and Vis AD, respectively) and infected (INFEC) or not (MOCK) with the SARS-CoV-2 ancient strain [CoV-2(B)] or the P.1 variant [CoV-2(P.1)]. **a** Overlap between differentially expressed proteins as determined by proteomics. Arcs represent different infection conditions (outside) or differentially expressed proteins (inside). Dark and light orange represent proteins that are differentially expressed in multiple conditions (linked by purple lines) or in only one condition, respectively. **b** Enriched ontology clusters. Statistically enriched terms were hierarchically clustered and the top-ranked terms within each cluster were displayed. **c** HSL abundance. Significant differences were determined by ANOVA and depicted as: a, $P < 0.01$ vs. MOCK CoV-2(B). **d** Western blotting of phosphorylated HSL (p-HSL), total HSL, ATGL, and α-tubulin as the loading control. Quantification represents the mean ± SEM of three independent pools from one donor. Significant differences were determined by a two-tailed Student's $t$-test: a, $P < 0.01$. **e** Glycerol

released in the medium. Significant differences were determined by a two-tailed Student's $t$-test: a, $P = 0.0018$. **f** SARS-CoV-2 genome in the supernatant of cells treated or not with 10 µM isoproterenol (ISO). Significant differences were determined by one-way ANOVA with Tukey's multiple comparison test and depicted as: a, $P < 0.0001$ vs. MOCK and b, $P < 0.05$ vs. INFEC[CoV-2(B)]. **g** Protein–protein interaction analysis using proteins related to the Reactome term "Interferon Signaling". **e**, **f** are mean ± SEM of two to six independent pools of cells from two to three donors. Proteomics were run using samples from two to three donors in technical triplicate. Different donors are distinguished by circles, squares, and triangles, while independent pools of cells from the same donor are marked by the same symbol. Source data are provided as a Source Data file and in Supplementary Data 1 and 2. The experiments were repeated once with similar results, with the exception of (**e**, **f**), which were not repeated; although the interassay variability was low, controls worked as expected, and biological and technical variability was accounted for in the analysis.

reduces viral titer in the supernatant of infected human adipose tissue cells[10]. These results suggest that inhibition of lipolysis in infected Sub AD cells is likely to interfere with SARS-CoV-2 replication. To test this hypothesis, we acutely stimulated Sub AD cell lipolysis for 30 min with the beta-adrenergic agonist isoproterenol and assessed whether an acute lipolytic stimulus increased the load of SARS-CoV-2 in the culture supernatant. Remarkably, we found that isoproterenol increased the levels of SARS-CoV-2 RNA in the conditioned medium of infected Sub AD cells by fourfold (Fig. 4f). Together, these results indicate that induction of lipolysis favors SARS-CoV-2 spreading in adipose tissue, which in turn suggests that the anti-lipolytic response triggered upon SARS-CoV-2 infection may represent an antiviral response.

### Opposing changes in the interferon signaling pathway in visceral fat cells by different SARS-CoV-2 lineages

In Vis AD cells, SARS-CoV-2 infection resulted in more specific alterations in pathways involved in vesicle trafficking and cellular responses to pathogens (Supplementary Fig. 7). When interrogating the Vis AD cell proteome for proteins involved in the "interferon signaling" pathway, we found seven proteins altered by infection with CoV-2(P.1), among which five were upregulated (i.e., IRF1, HLA-C, TPR, EIF4G1, and TRIM28) (Fig. 4g). In contrast, we found six proteins altered by infection with CoV-2(B), among which five were downregulated (i.e., STAT1, HLA-E, TPR, TRIM25, and TRIM28) (Fig. 4g). These results show that different SARS-CoV-2 lineages may elicit different antiviral responses in visceral adipose tissue cells. Together with the viral load data, these results suggest that enhanced interferon signaling may help limit the replication and spread of a SARS-CoV-2 VOC in visceral adipose tissue. In contrast, inhibition of interferon signaling may favor productive infection in cells exposed to the CoV-2(B) lineage.

### Higher induction of proinflammatory gene expression in visceral versus subcutaneous fat cells infected with SARS-CoV-2

Adipose tissue-derived cytokines may act locally and systemically to set the inflammatory tonus of the organism while contributing to the onset of metabolic diseases[32]. Considering that adipose tissue cell infection with SARS-CoV-2 resulted in alterations in several pathways involved in inflammatory processes, we investigated the expression of genes encoding key cytokines that play a role in metabolic dysfunction and COVID-19 pathogenesis. The baseline levels of IFNA and IL6 were higher in Vis AD vs. Sub AD cells (Fig. 5a, c, g, i). IL1B level was lower in Vis AD cells (Fig. 5d, j), and there were no baseline statistical differences for IFNB, TNFA, and CCL2 expression between Vis AD and Sub AD cells (Fig. 5b, e, f, h, k, l). In Sub AD cells, SARS-CoV-2 did not alter cytokine expression except for a modest increase in IL1B and CCL2—the latter only observed after infection with the CoV-2(P.1) variant (Fig. 5d, f, j, l). In contrast, marked changes occurred in Vis AD cells, where CoV-2(B) infection led to upregulation of most cytokines, including TNFA, which was increased by over 100-fold (Fig. 5e). In agreement with the viral load and proteomics data, the cytokine response after CoV-2(P.1) exposure was attenuated (Fig. 5g–l).

## Discussion

Our results provide a mechanistic framework to help explain the exacerbated metabolic and inflammatory symptoms manifested by COVID-19 patients (Fig. 6). Early reports characterized hyperglycemia and poorly controlled blood glucose levels as risk factors of severe illness by COVID-19[4,9,33,34]. Moreover, normoglycemic individuals can develop a type of acute, new-onset diabetes when infected with SARS-CoV-2, which poses an even greater risk of morbidity and mortality by COVID-19[35–37]. While hyperglycemia in COVID-19 has been associated with reduced adiponectin levels[9], the causes of COVID-19-induced diabetes are likely to be multifactorial[33]. Confirming a possible role for adiponectin in COVID-19-related metabolic dysfunction, we observed a trend toward downregulation of adiponectin protein levels in subcutaneous adipose tissue cells infected with CoV-2(B) (q value = 0.06; Supplementary Data 1). We also saw upregulation of genes encoding proinflammatory cytokines in visceral adipose tissue cells infected by SARS-CoV-2 [mainly the CoV-2(B) lineage], unveiling yet another potential underlying contributor to metabolic dysfunction in COVID-

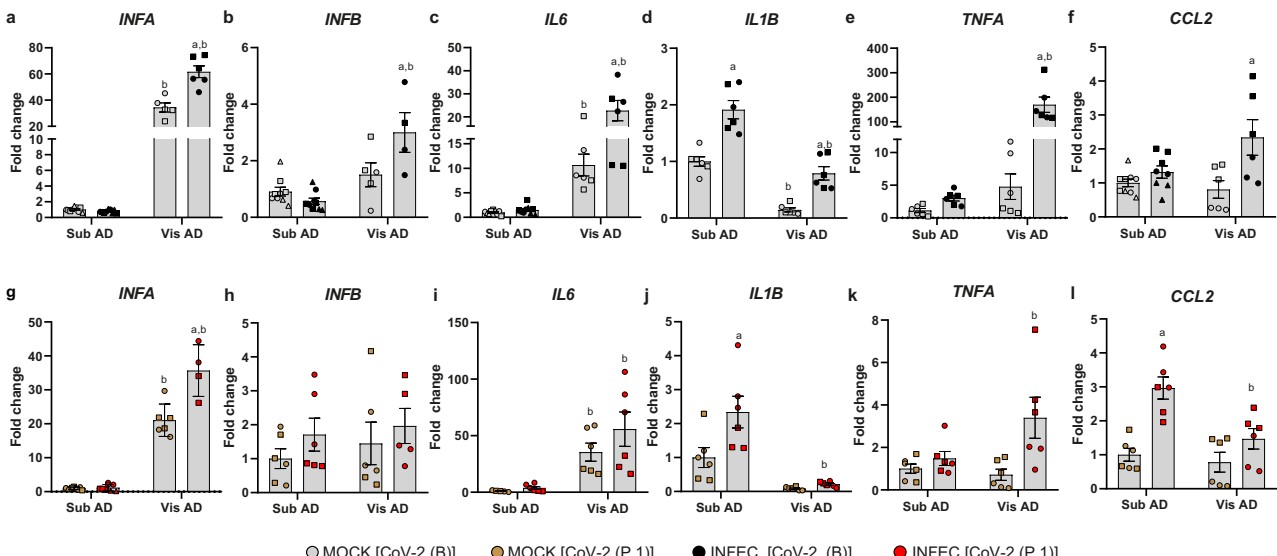

**Fig. 5 | Genes associated with inflammation are differentially expressed in SARS-CoV-2-infected adipose tissue cells. a–l** Stromal-vascular cells from human subcutaneous and visceral fat depots were differentiated into adipocytes (Sub AD and Vis AD, respectively) and infected (INFEC) or not (MOCK) with the SARS-CoV-2 ancient strain [CoV-2 (B)] (**a–f**) or the P.1 variant [CoV-2 (P.1)] (**g–l**). Gene expression as quantified by RT-qPCR 24 h post-infection. Two-way ANOVA with Tukey's multiple comparison test was applied to identify differences between groups. These differences are depicted in letters: a, $P < 0.05$ vs. respective MOCK; b, $P < 0.05$ vs. respective Sub AD. Data represent the mean ± SEM of two to three independent pools of cells from two to three donors. Different donors are distinguished by circles, squares, and triangles, while independent pools of cells from the same donor are marked by the same symbol. Source data are provided as a Source Data file. These experiments were not repeated; although the calculated interassay variability for these genes was low, controls worked as expected, and biological and technical variability was accounted for in the analysis.

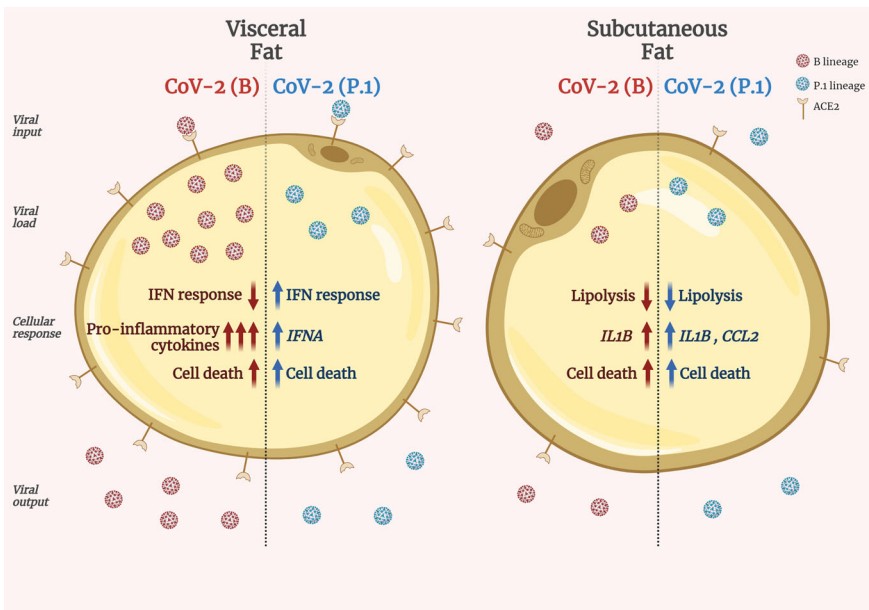

**Fig. 6 | SARS-CoV-2 infects adipose tissue in a fat depot- and viral lineage-dependent manner.** CoV-2(B), original B SARS-CoV-2 lineage. CoV-2(P.1), SARS-CoV-2 gamma variant P.1. IFN interferon, IFNA interferon alpha gene, IL1B interleukin 1 beta gene, CCL2 C-C motif chemokine ligand 2 gene. Created with Biorender.com.

19 patients. Indeed, visceral adipose tissue inflammation is one of the main determinants of type 2 diabetes in individuals with obesity[38,39]. This is partly because the liver is in direct contact with visceral fat through the portal vein, which causes proinflammatory cytokines produced in the visceral adipose tissue to reach the liver at high levels and induce hepatic insulin resistance. In addition, hyperglycemia promotes SARS-CoV-2 replication and increases cytokine production in macrophages, as we have previously demonstrated[40], further enhancing systemic inflammation. In parallel, SARS-CoV-2 infection leads to inhibition of lipolysis in subcutaneous adipose tissue. The systemic impact of such inhibition remains to be determined, considering that reports are conflicting regarding the levels of free fatty acids in COVID-19 patients[41–43].

Although our in vitro model does not allow for a certain distinction of the cell types serving as the primary sites for SARS-CoV-2 infection and replication in adipose tissue, we speculate that adipocytes are likely a major site. We conclude that based on several reasons. Firstly, we found adipocytes infected by SARS-CoV-2 in the adipose tissue of COVID-19 patients. Moreover, whereas there are considerable amounts of undifferentiated preadipocytes in our cell cultures, the culture is enriched with differentiated adipocytes and, as demonstrated by ref. 10 and ref. 8, undifferentiated preadipocytes are not infected by SARS-CoV-2. Indeed, we found no infectious particles in the supernatant of confluent, undifferentiated human adipose tissue-derived stromal-vascular cells 24 h after infection with SARS-CoV-2. Moreover, we observed viral particles in differentiated cells harboring lipid droplets. These results are consistent with recent studies demonstrating that SARS-CoV-2 replication is favored in lipid-laden cells[5,10], further indicating that adipocytes are propitious sites for SARS-CoV-2 infection and propagation. Macrophages are yet another potential site for SARS-CoV-2 replication in adipose tissue, although our in vitro model did not include these cells in relevant abundance. In a recent report, adipose tissue macrophages have also been found to bear SARS-CoV-2 and, along with infected adipocytes, promote adipose tissue inflammation and contribute to tissue dysfunction[8].

In addition to confirming the potential of SARS-CoV-2 to infect adipose tissue cells, here we demonstrate that visceral fat, which is more abundant in the population at greater risk for the severe illness from COVID-19[15–18], has cells that are more prone to SARS-CoV-2 infection and respond differently to the virus when compared to its subcutaneous counterpart. We also show that, although significant cell death is observed in cells from both depots in response to SARS-CoV-2 infection, which could further exacerbate tissue dysfunction and inflammation and induce adipose tissue remodeling[11], visceral fat cells are intrinsically more susceptible to the viral infection, potentially serving as a more critical virus reservoir in COVID-19 patients. Finally, we demonstrate that visceral adipose tissue cell infection by the ancient SARS-CoV-2 lineage leads to the downregulation of proteins involved in the interferon signaling pathway despite a pronounced induction of proinflammatory markers. This is in sharp contrast with the response of visceral fat cells to the infection with the gamma variant, which leads to the upregulation of proteins involved in the interferon signaling pathway and results in a much milder induction of proinflammatory markers. This finding corroborates the relevance of visceral adipose tissue infection in COVID-19 pathophysiology, considering that attenuation of viral pathogenicity is a common phenomenon in light of evolution[44]. In future studies, it would be interesting to compare the behavior of adipose tissue SARS-CoV-2 infection to other viruses that have been shown to infect fat cells (e.g., influenza A virus, human immunodeficiency virus, adenovirus)[45–47], as well as to investigate whether the way obesity influences the pathogenesis of infectious diseases other than COVID-19 (e.g., influenza)[48,49] is also linked to the abundance of visceral vs. subcutaneous fat or the differential susceptibility to infection these fat depots may exhibit.

In summary, based on the data presented hitherto, we conclude that adipose tissue infection represents a relevant feature of COVID-19 and should be considered in all its complexity when evaluating the impact of different VOCs on disease pathogenesis and symptomatology.

## Methods
### Materials
Materials/reagents are listed with catalog numbers and vendors in Supplementary Data 3. Unique materials are available from authors upon reasonable request pending donor's and/or ethics committee consent if necessary.

## Human subjects

The protocols for human sample collection, storage, and analysis were conducted according to the guidelines of the Declaration of Helsinki and approved by the Institutional Research Ethics Committees of the University of São Paulo and University of Campinas (CAAE 48836721.3.0000.5440 and CAAE 78577417.8.0000.5404). The latter was confirmed by the National Ethics and Research Council. Procedures for sample collection, processing, and subject information are described below. Donors were not compensated.

## Postmortem sample collection

Minimally modified invasive autopsy (MIA)[50] was performed on 47 patients who died from COVID-19 at the Clinics Hospital of Ribeirão Preto Medical School, University of São Paulo, Ribeirão Preto, SP, Brazil—from May to July 2020. Characteristics of these patients are described in Supplementary Table 1 and elsewhere[51]. Briefly, all MIAs were performed at the bedside through postmortem surgical thoracic subcutaneous adipose tissue biopsy within 1 h of death by a 3-cm incision on the anterior side of the chest between the fourth and fifth ribs. A matching 14-gauge cutting needle (Magnum Needles, Bard Care) and a biopsy gun (Magnum, Bard Care) were used. Written informed consent was waived upon approval by the Institutional Research Ethics Committees of the University of São Paulo following the guidelines of the National Ethics and Research Council and according to its resolution number 466 of December 12, 2012, item IV.8.

## Human adipose tissue stromal-vascular cell isolation and culture

We isolated adipose tissue-derived stromal-vascular cells from abdominal subcutaneous adipose tissue and visceral omental adipose tissue of three individuals who underwent abdominal surgery (i.e., bariatric surgery or cholecystectomy) at the Clinics Hospital of the University of Campinas, Campinas, SP, Brazil. These cells had been isolated prior to the COVID-19 pandemic. Hence, donors were not infected with SARS-CoV-2. The subjects received written and oral information before giving written informed consent to collect the biopsy and use the tissues. Cells used in this study were obtained from 3 donors (one male that underwent cholecystectomy and two females that underwent bariatric surgery) aged 33, 43, and 46 (mean of 41 years of age), with BMI of 26, 34, and 35 (mean of 32). Donors were not diabetic. One of them had nonalcoholic steatohepatitis. All material used in the procedure was sterile. The biopsies were collected during surgery and transported to the laboratory in sealed, sterile falcon tubes to initiate the procedure of stromal-vascular fraction isolation within 30 min. The tissue was added to a petri dish, dissected to remove the connective/fibrotic tissue, and then weighed. The fat tissue was cut into small pieces and then digested with 25–30 ml of lysis buffer [1 mg/ml collagenase type 2 (Worthington) in Hanks' Balanced Salt Solution (HBSS) containing 2% bovine albumin serum (BSA)—essentially fatty acid-free (Sigma-Aldrich, A6003), filtered through 0.22 μm] at 37 °C for 30–50 min with slight agitation or until homogeneous. The homogenate was then filtered through a 250 μm filter and collected in a sterile falcon tube. After a rest (~5 min), the infranatant containing the stromal-vascular fraction was collected using a Pasteur pipette and centrifuged for 5 min at $200 \times g$ and 4 °C. The supernatant was discarded, and the cell pellet was washed with HBSS. The procedure of centrifugation and washing was repeated twice, and the pellet was washed once with a Red Blood Cell lysis buffer. The pellet was then resuspended in BM-1 medium (ZenBio) supplemented with 10% fetal bovine serum (FBS) and 1% penicillin/streptomycin (penicillin 10,000 units/ml, streptomycin 10 mg/ml). Cells were cultured at 37 °C and 5% $CO_2$ atmosphere until semi-confluency (80-90%). They were then trypsinized and frozen in a freezing medium [10% dimethyl sulfoxide (DMSO), 50% FBS, and 40% BM-1] and stored in a liquid nitrogen biorepository for further analysis (e.g., adipocyte differentiation followed by exposure with agents, lipid quantification, determination of cell viability, gene expression analysis, proteomics, immunodetection of proteins, quantification of cell's secretion products, and metabolic analysis).

## Adipocyte differentiation

Differentiation of human adipose tissue-derived stromal-vascular cells into adipocytes was performed according to a protocol published elsewhere with slight modifications[27]. Adipose tissue-derived stromal-vascular cells were thawed in BM-1 medium supplemented with 10% FBS, 1% penicillin/streptomycin, 17 ng/ml FGF-Basic (bFGF), and 15 ng/ml bone morphogenetic protein 4 (BMP4) and cultured at 37 °C and 5% $CO_2$ atmosphere. Cells were expanded (i.e., split one to three times) and then seeded onto 24-well plates at a density of 40,000 cells/cm$^2$ for adipocyte differentiation. When cells reached 100% confluency (day 0), medium was replaced by BM-1 supplemented with 3% FBS, 1% penicillin/streptomycin, 0.1 μM dexamethasone, 500 μM 3-isobutyl-1-methylxanthine (IBMX), 20 nM insulin, 5 nM triiodothyronine (T3), and 10 ng/ml BMP4. Cells were cultured for 7 days, and the medium was changed every 2–3 days. On day 7, the medium was replaced by BM-1 supplemented with 3% FBS, 1% penicillin/streptomycin, 0.1 μM dexamethasone, 20 nM insulin, 5 nM T3, and 10 ng/ml BMP4 until the cells were fully differentiated at day 10. Cells were then cultured for an additional 3 days with BM-1 supplemented with 3% FBS and 1% penicillin/streptomycin and subjected to the experimental procedures described below.

## Oil red O staining and quantification

Cells were washed with phosphate-buffered saline (PBS) and fixed with 4% formaldehyde in PBS for 15 min at room temperature. Cells were then washed with PBS and stained with filtered oil red O solution (5 g/L in isopropanol) diluted 3:2 in water for 30 min at room temperature. Cells were then washed five times with distilled water. For oil red O quantification, we extracted the dye incubating stained cells with 100% isopropanol for 10 min at room temperature on an orbital shaker and measured absorbance at 490 nm in the eluate in duplicate.

## Viruses and cell lines

The ancient SARS-CoV-2 viral lineage [CoV-2(B)] was isolated from the second confirmed case of COVID-19 in Brazil (SARS.CoV-2/SP02.2020, GenBank accession number MT126808)[40]. The SARS-CoV-2 gamma variant [CoV-2(P.1)] was isolated from residual nasopharyngeal lavage specimens of a patient from Manaus City, Brazil, who tested positive for SARS-CoV-2 (GISAID Accession ID EPI_ISL_1708318)[52]. Viral stocks were propagated in Vero CCL-81 cells (ATCC, CCL-81) cultured in Dulbecco's Modified Eagle Medium (DMEM) supplemented with 10% FBS and 1% of penicillin/streptomycin at 37 °C with 5% $CO_2$ atmosphere. The viral titer was determined by the plaque assay (see below).

## Generation of monoclonal antibodies against NRP1 b1b2

Female BALB/c and C57BL/6 mice, 8–9 weeks old, were immunized intraperitoneally with 17 μg of recombinant NRP1 b1b2 mixed with an equal volume of complete Freund's adjuvant (Sigma Aldrich), followed by a booster immunization four weeks later of the same dose mixed with incomplete Freund's adjuvant (Sigma Aldrich). The mice received three boosts of the same amount of antigen in PBS on days −3, −2, and −1 prior to fusion. The spleens were excised, and the splenocytes were fused with myeloma cells (P3X63Ag8.653, obtained from ATCC, CRL-1580) according to the previously described protocol[53]. Beginning on day 10 after fusion, hybridoma supernatants were screened for specific antibodies. Before experiments, the hybridoma supernatants were centrifuged at $300 \times g$ for 5 min at room temperature, and 500 μL were dialyzed against 2 L of PBS overnight at 4 °C.

## Infection and harvesting

Cells were incubated with the CoV-2(B) or CoV-2(P.1) lineage (MOI = 1 unless indicated otherwise) for 1 h at room temperature under gentle agitation (20 RPM) on an orbital shaker in a biosafety level 3 laboratory (BSL3). After viral adsorption, cells were washed once with PBS and incubated with a culture medium under standard culture conditions (37 °C and 5% $CO_2$ atmosphere) for the period indicated in the figure legend. For NRP1 neutralization, cells were incubated for 6 h prior to the infection and 1 h during the infection with the neutralizing antibodies anti-neuropilin-1 3E7 (2.6 µg/ml) or 3E8 (2.3 µg/ml) and/or the selective NRP1 antagonist EG00229 (10 mM)[29,30,54]. Anti-neuropilin-1 antibody specificity and ability to prevent SARS-CoV-2 infection have been previously demonstrated[29,30].

## Cell viability assay

Cell viability was measured using a CellTiter-Glo luminescent cell viability assay (Promega) according to the manufacturer's instructions. Cell viability was measured on days 1, 2, 3, 4, and 5 after infection.

## Plaque assay

Standard plaque assay was performed using Vero cells. Cells were seeded onto 24-well plates ($4 \times 10^4$ cells per well) and incubated with tenfold serial dilutions (up to $10^{-5}$ dilutions) of isolated viruses or supernatants of differentiated cells for 1 h at room temperature under standard culture conditions. The culture medium was replaced by 1 ml of an overlay medium containing DMEM, 1% carboxymethyl cellulose (CMC), 5% FBS, and 1% penicillin/streptomycin. Four days later, cells were fixed with 8% formaldehyde in PBS for 24 h at room temperature and stained with 1% w/v methylene blue solution. Cells were then washed with water to reveal the plaques. The results are expressed as plaque-forming units (PFU)/ml with a limit of detection of 400 PFU/ml.

## RNA extraction

**Human biopsies.** Paraffin-embedded tissues (ten 10-µm sections) were deparaffinized with xylene and washed with absolute ethanol. RNA was extracted by overnight incubation with lysis buffer (1 M Tris pH 8.0; 0.5 M EDTA pH 8; 10% SDS) and proteinase K (1 U/350 µl, Life Technologies) at 55 °C, and after this process, Trizol reagent (Thermo Fisher Scientific) was added to the mixture. Total RNA was extracted according to the manufacturer's instructions.

**Cells.** RNA was extracted from differentiated adipose tissue-derived cells and Vero cells plated in 24-well plates. Wells were washed with PBS, and cells were immediately harvested with 300 µl of Trizol or incubated for 3 min with 100 µl trypsin-EDTA solution (0.25%, Thermo Fisher Scientific) at 37 °C. Trypsin was neutralized with 500 µl of BM-1 medium supplemented with 10% FBS and cells were centrifuged at 250 × g for 5 min. The supernatant was removed, and the pellet was resuspended in 500 µl PBS and centrifuged again at 250 × g for 5 min. The supernatant was discarded, and 300 µl of Trizol were added to the pellet. Total RNA was extracted according to the manufacturer's instructions.

**Medium.** Twenty-four hours after infection, differentiated cells were serum-starved for 2 h and incubated for 30 min with Krebs-Ringer-HEPES buffer (KRH buffer: 120 mM NaCl; 4.7 mM KCl; 2.2 mM $CaCl_2$; 1.2 mM $KH_2PO_4$; 1.2 mM $MgSO_4$; 5.4 mM glucose; 10 mM HEPES pH 7.4) supplemented with 4% BSA without fatty acid (A7030, Sigma-Aldrich) containing 10 µM isoproterenol or vehicle. The medium was collected and diluted in Trizol (1:10). Total RNA was extracted according to the manufacturer's instructions.

## Viral load and gene expression analysis

SARS-CoV-2 RNA was quantified using RT-qPCR with primer-probe sets for N2 or E genes, according to US Centers for Disease Control and Prevention and Charité group protocols (Supplementary Table 2)[55]. Real-time qPCR assays were conducted on a Step-One Plus thermocycler (Applied Biosystems) or in a CFX384 Touch Real-Time PCR Detection System (Bio-Rad). In the case of postmortem human adipose tissue biopsies, we performed amplification of the N2 gene and virus quantification was normalized by 1 million copies of RNase P. One microgram of RNA was used for genome amplification with 20 µM specific primers/probes and 5 µM qPCRBIO Probe 1-Step Go (PCR Biosystems) with the following parameters: 3 min at 95 °C, 45 cycles at 95 °C for 15 s and 60 °C for 30 s, and holding at 10 °C. Genome loads of SARS-CoV-2 were determined using a standard curve prepared with a plasmid containing a 944 bp amplicon of the N gene, inserted into a TA cloning vector (PTZ57R/T) using the CloneJet Cloning Kit (Thermo Fisher Scientific). In the case of adipose tissue-derived cells, we quantified the SARS-CoV-2 E gene. About 500 ng of RNA was converted into cDNA by reverse transcription reaction using random hexamers according to the manufacturer's instructions (High-Capacity cDNA Reverse Transcription Kit, Applied Biosystems). The amplification was done in a reaction containing: 10 ng of cDNA, 10 µM primers, 5 µM probe, and qPCRBIO Probe Mix (PCR Biosystems). The cycling condition was: 95 °C for 2 min (1 cycle), 95 °C for 5 s, and 63 °C for 30 s (40 cycles) in the QuantStudio 3 (Applied Biosystem). The absolute quantification of the viral genome was determined by a standard curve made by the titration of the SARS-CoV-2 genome quantified as equivalent to focus forming units (eFFU) and normalized by total RNA. We extracted total RNA and synthesized cDNA for gene expression analysis as mentioned above and performed PCR quantification using 15 ng of cDNA, 300 nM of specific primers (Supplementary Table 2), and QuantiNova SYBR Green (Qiagen). Amplification occurred upon cycling conditions: 95 °C for 3 min and 39 cycles of 95 °C for 15 s, 60 °C for 20 s, and 72 °C for 30 s. Amplification plots, melting curves, and Ct values were obtained using CFX Manager (Bio-Rad). Relative gene expression was calculated using 2-ΔΔCt. We used *GAPDH* as the housekeeping gene as the expression of this gene does not change in adipose tissue cells after the infection with SARS-CoV-2.

## Immunofluorescence

**Human biopsies.** Tissue sections of paraffin-embedded biopsies obtained from COVID-19 fatal cases were deparaffinized, heated to 60 °C for paraffin melting, washed in xylenes baths, and rehydrated in graded ethanol followed by deionized water. Antigen retrieval was done by incubating slides in buffer Tween 20-Citrate (10 mM sodium citrate, 0.05% Tween 20, and pH 6.0) in microwaved pressure cooked for 10 min at 70% wattage power. After 1 h at room temperature, tissue sections were blocked with superblock (SuperBlock Blocking Buffer in PBS, Pierce), followed by overnight incubation with the primary antibodies for anti-SARS-CoV-2 Spike (Rheabiotech, IM-0828, 1:100), perilipin 2 (Anti-ADFP, Abcam, ab78920, 1:100) or perilipin 1 (G-2) (Santa Cruz Biotechnology, sc-390169, 1:50) at 4 °C in a humidified chamber. Slides were washed six times in PBS, incubated with the secondary antibodies conjugated with Alexa Fluor 488 (Thermo Fisher Scientific, A21202, 1:800), 594 (Abcam, ab150116, 1:800), or 405 (Abcam, ab175652, 1:800). Nucleus staining was carried out using 4′,6-diamidino-2-phenylindole dihydrochloride dye (DAPI, Thermo Fisher cat. 62248) for 1 h at room temperature, washed three times in PBS, and mounted with ProLong® Gold Antifade Mounting (Thermo Fisher Scientific). Slides were examined in a TCS SP5 confocal microscope (Leica Microsystems) and analyzed with ImageJ/Fiji software. For the anti-SARS-CoV-2 spike, negative controls were Vero CCL-81 mock slides and samples from non-infected individuals, and positive controls were SARS-CoV-2-infected Vero CCL-81 cell slides treated the same way, omitting antigen retrieval incubation.

**Cells.** Cells were fixed with 4% paraformaldehyde and blocked with 1% BSA (Sigma-Aldrich) in PBS at room temperature for three times. For

labeling of intracellular antigens, a permeabilization step was performed when samples were incubated for 10 min with PBS containing 0.2% Tween 20 or only PBS for labeling membrane-associated antigens. To block nonspecific binding of antibodies, cells were incubated with 1% BSA, 22.52 mg/mL glycine in PBS-T (PBS + 0.1% Tween 20) or 1% BSA, 22.52 mg/mL glycine in PBS for 30 min. Samples were incubated overnight at 4 °C with the primary antibodies: rabbit monoclonal anti-SARS-CoV-2 Spike (Invitrogen, 703959, 1:500) and mouse monoclonal anti-double-stranded RNA J2 (dsRNA, SCICONS, 10010200, 1:1,000), or polyclonal goat anti-ACE2 (R&D systems, AF933, 1:200). The slides were washed three times with PBS-T (PBS with 22.52 mg/mL Glycine; 0.01% Tween 20; 1% BSA) and incubated at room temperature for 1 h with the secondary antibodies: alpaca anti-mouse IgG Alexa Fluor 488 (Jackson ImmunoReseacher, 615-545-214, 1:1,000), goat anti-rabbit IgG Alexa Fluor 405 (Abcam, ab175652, 1:800), or donkey anti-goat IgG Alexa Fluor 488 (Abcam, ab150129, 1:800). Lipid droplets were stained with HCS LipidTOX™ Deep Red (Invitrogen, H34477, 1:200) and morphology was assessed by the bright field. Slides were then mounted with VECTASHIELD Antifade Mounting Medium with DAPI (Vector Laboratories, H-1200-10) or ProLong Gold Antifade Mountant (Invitrogen, P36930). Images were acquired by Axio Observer combined with an LSM 780 confocal microscope (Carl Zeiss) at 63X magnification at the same setup (zoom, laser rate). Images were analyzed with Fiji by ImageJ.

## Sample preparation, liquid chromatography-mass spectrometry, and data processing for proteomics

The proteome of differentiated adipose tissue-derived cells was analyzed by liquid chromatography-tandem mass spectrometry (LC-MS/MS) 24 hpi with SARS-CoV-2. Biological duplicates or triplicates for each condition were processed and analyzed in technical triplicate. Cells were chemically lysed with Lysis Buffer (100 mM Tris-HCl, 1 mM EDTA, 150 mM NaCl, 1% Triton-X-100, protease, and phosphatase inhibitors) and ultrasonication (three cycles of 20 s each with 90% of frequency). Protein extract was quantified by Pierce BCA kit according to the manufacturer's instructions (Thermo Fisher Scientific). About 20 µg of the protein extract was transferred to a Microcon-10 Centrifugal Filter, with a 10 kDa cutoff, for FASP protein digestion[56]. Proteins were reduced (10 mM DTT), alkylated (50 mM IAA), and digested overnight by trypsin at 37 °C in 50 mM ammonium bicarbonate (AmBic), pH 8.0. On the following day, the peptides were recovered from the filter in 50 mM AmBic, and trypsin activity was quenched by adding formic acid (FA) to a final concentration of 1% (v/v). After that the peptides were concentrated in SpeedVac and stored at −80 °C until use. Digested peptides were resuspended in 0.1% FA. LC-MS/MS analysis was performed in an ACQUITY M-Class System (Waters Corporation) coupled online to a Synapt G2-Si mass spectrometer (Waters Corporation). One microgram of peptides were loaded onto a trapping column (Symmetry C18 5 µm, 180 µm × 20 mm, Waters Corporation) and subsequently separated in the analytical column (HSS T3 C18 1.8 µm, 75 µm × 150 mm; Waters Corporation). For gradient elution, 0.1% FA was used as eluent A and acetonitrile-FA (99.9% ACN:0.1% FA) as eluent B. Reversed-phase gradient was carried out over 120 min, with a linear gradient 3−60% acetonitrile over 60 min at 300 nL/min. In the Synapt G2-Si, the peptide spectra were acquired by ion mobility-enhanced data-independent acquisition (HDMSe). Mass spectrometry analysis was performed in "Resolution Mode", switching between low (4 eV) and high (25−60 eV) collision energies, using a scan time of 1.0 s per function over 50−2000 m/z. The wave velocity for ion mobility separation was 1000 m/s and the transfer wave velocity was 175 m/s. A [Glu1]-Fibrinopeptide B Standard (Waters Corporation) was used as the reference lock mass compound. The raw data from each experiment was processed in Progenesis QI for proteomics (Waters Corporation). Tandem mass spectra were searched against the Uniprot *Homo sapiens* reviewed proteome database (20376 entries, released 2022-03), using tolerance parameters of 20 ppm for precursor ions and 10 ppm for product ions. For peptide identification, carbamidomethylation of cysteines was set as a fixed modification, oxidation of methionines as a variable modification, two missed cleavages were permitted, and a false discovery rate (FDR) was capped at 1%. Protein identification was performed using a minimum of one fragment ion matched per peptide, a minimum of three fragment ions per protein, and a minimum of two peptides per protein. Label-free quantitative analysis was carried out using the relative abundance intensity normalized by all peptides identified. The expression analysis was performed considering the technical replicates for each experimental condition, following the hypothesis that each group is independent. Proteins with ANOVA ($p$) corrected for multiple testing by an optimized FDR approach ($q$) ≤0.05 between groups were considered differentially expressed.

## Protein ontologies and network analysis

Proteins differentially expressed ($p < 0.05$) were submitted to systems biology analysis in R (v. 4.0.3) and Cytoscape environments[57]. Proteins were enriched using ClusterProfiler R package[58] and Kyoto Encyclopedia of Genes and Genomes (KEGG)[59]. For the network analysis, we used Reactome[60] for module detection and enrichment pathway analysis. For overlap analysis and enriched ontology clusters, we used Metascape[61].

## Re-analysis of single-nucleus RNA sequencing data

Single-nucleus RNA sequencing (snRNASeq) from human brown adipose tissue was retrieved at https://www.ebi.ac.uk/arrayexpress/experiments/E-MTAB-8564/[28]. Raw matrices generated by Cellranger were downloaded and filtered to keep drops with at least 1000 counts. Seurat 4 package was used for single-nucleus analysis and annotation of cell clusters. Seurat object was created filtering for genes detected in at least ten cells and cells with at least 200 genes detected. The SCT workflow from Seurat was used with default settings and a resolution of 0.4 for cluster identification. Cell clusters were annotated according to the following markers: conventional T (Tconv) cells (PTPRC+ CD4+ FOXP3−), regulatory T (Treg) cells (PTPRC+ CD4+ FOXP3+ IL2RA+), T CD8+ cells (PTPRC+ CD8+), adipocytes (ADIPOQ+ PLIN1+), preadipocytes (CD34+), fibroblasts (VEGFC+ VCL+), smooth muscle cells (MYH11+), endothelial cells (PECAM1+ VWF+), B cells (CD19+), macrophages (CD163+), and macrophage/myeloid cells (CD163+ IL2RA+). Clusters not following these criteria were annotated as "uncharacterized".

## Western blotting

Cells were washed with PBS and lysed in RIPA buffer (Cell Signaling Technology) containing a protease and phosphatase inhibitor cocktail (Thermo Fisher Scientific). Cell lysates were sonicated and centrifuged at $14,000 \times g$ for 20 min at 4 °C. Protein concentration was determined in the supernatants using the Pierce BCA kit (Thermo Fisher Scientific). Lysates were then denatured in Laemmli buffer (0.5 M Tris, 30% glycerol, 10% SDS, 0.6 M DTT, 0.012% bromophenol blue) and heated at 95 °C for 5 min. Proteins (15 to 20 µg) were loaded onto the gel and resolved by 8% SDS-polyacrylamide gel electrophoresis (SDS-PAGE). Proteins were transferred to nitrocellulose membranes (Bio-Rad) and membranes were blocked for 1 h at room temperature in Tris-buffered saline (137 mM NaCl, 20 mM Tris-HCl, pH 7.6) containing 0.1% Tween 20 (TBS-T) and 5% low-fat milk or StartingBlock™ T20 (PBS) Blocking Buffer (Thermo Scientific), followed by incubation with the primary antibody overnight at 4 °C. Primary antibodies used were produced in rabbits−HSL (Cell Signaling Technology, 4107, 1:1,000), phospho-HSL(S565) (Cell Signaling Technology, 4137, 1:1,000), ATGL (Cell Signaling Technology, 2138, 1:1,000), and α-tubulin (Cell Signaling Technology, 2144, 1:5,000)−or mice−ACE-2 (Novus Biological, NBP2-80035, 1:3,000) and Vinculin (Abcam, ab207440, 1:2,000). Protein-bound antibodies were detected with a secondary, HRP-linked anti-

rabbit (Cell Signaling Technology, 7074, 1:10,000) or HRP-linked anti-mouse (Cell Signaling Technology, 7076, 1:10,000) antibody and visualized by chemiluminescence using Clarity Western ECL Substrate (Bio-Rad). To quantify band density, identically sized regions of interest were drawn on each lane and pixel density was measured using the ImageJ software. The bands of interest and their quantification are presented in the main figures, while the uncropped and unprocessed scans are included in the Source Data file.

### Glycerol assay (Lipolysis)

After 24 h post-infection, cells were serum-starved for 2 h. Cells were then incubated for 30 min with a KRH buffer supplemented with 4% BSA, fatty acid-free (Sigma Aldrich, A7030). The medium was collected, stored on ice for 10 min, and placed in a dry bath at 60 °C for 20 min. Glycerol was quantified in the medium using the 3T3-L1 Lipolysis Assay Kit according to the manufacturer's instructions (Zenbio). Results were normalized by the amount of protein available in each well as quantified by the BCA kit (Thermo Fisher Scientific).

### Statistics

Results are presented as mean ± SEM. Statistical analyses were performed using Graph Prism 8.0 or the platforms used for proteomic data processing as described above. For the postmortem tissue analysis, each data point represents a sample from one individual. We used a two-tailed Student's $t$-test for two group comparisons or one-way ANOVA for three group comparisons. We used Pearson correlation analysis to test the correlation between two variables. For the in vitro studies, dot plots represent experimental replicates (i.e., independent pools of cells from the same donor) from one to three biological samples (i.e., different donors), as indicated in the figure legends. Comparisons were made by accounting for the effect of the infection on each donor (considering the variability among different pools of cells from the same donor) and among different donors (considering the interindividual variability). Depending on the number of variables, we applied one-, two-, or three-way ANOVA in combination with multiple comparison tests (Tukey's or Sidak's) as indicated in the figure legends to determine the effect of infection, time, cell origin, and/or viral lineage and the interaction between these variables. We used the Shapiro–Wilk test to examine if the variables were normally distributed. Only on two occasions when the sample size was too small to establish normality (Supplementary Fig. 2d, e), we assumed normal distribution. A confidence interval of 95% was used and the alternative hypothesis was accepted when $P < 0.05$. We depicted statistically significant differences in the graphs by using letters and $P$ values are displayed in the figure legend and Source Data file.

### Reporting summary

Further information on research design is available in the Nature Research Reporting Summary linked to this article.

## Data availability

All data generated or analyzed during this study are included in this published article (and its supplementary information files) or in public repositories. The proteomics data have been deposited at the ProteomeXchange Consortium via the PRIDE partner repository with the dataset identifier PXD034510 and DOI: 10.6019/PXD034510. Uniprot *Homo sapiens* reviewed proteome database (released 2022-03) was used for tandem mass spectra analysis. Single-nucleus RNA sequencing analysis in this study re-used previously published[28] and publicly available data deposited in the ArrayExpress database with the dataset identifier E-MTAB-8564. Donor information is presented in Supplementary Table 1 as averages to avoid individual identification. Individual data are protected and are not available due to data privacy laws. Source data are provided with this paper.

## Code availability

This paper does not report original code.

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

## Acknowledgements

We thank Elzira E. Saviani, Silvia R. Nascimento, and Roberta R. C. Rosales for technical support, Edson Durigon for providing the ancient SARS-CoV-2 lineage, and Laís Coimbra and Alexandre Borin for assistance in the production of viral stocks. We acknowledge the São Paulo Research Foundation (FAPESP) (2017/08264-8 and 2020/05040-4 to L.O.L., 2020/04746-0 and 2019/00098-7 to D.M.-d.-S., 2019/05155-9 to V.C.C., 2020/04583-4 to M.A.R.V., 2020/15959-5 to M.K.O., 2019/26119-0 to E.A., 2016/00194-8 and 2020/04558-0 to J.L.P.-M., 2020/04579-7 to P.M.M.M.-V., 2021/10373-5 to F.M.-N., 2020/08716-9 to T.D.S., 2013/07607-8, 2020/04919-2, and 2017/01184-9 to M.A.M., 2017/23920-9 to R.G.L., 2016/24163-4 to G.S.P., 2019/04726-2 to T.L.K., and 2018/21635-8 to G.P.R.), Fundo de Apoio ao Ensino, Pesquisa e Extensão (FAEPEX) Unicamp (2266/20 to J.L.P.-M., 2456/20 and 2274/20 to M.A.M.), the Brazilian National Council for Scientific and Technological Development (CNPq) (403201/2020-9 and 310100/2017-8 to E.A., 305628/2020-8 to J.L.P.-M., and 310287/2018-9 to M.A.M.), the Coordenação de Aperfeiçoamento de Pessoal de Nível Superior (CAPES)—Finance Code 001 (88887.356527/2019-00 to M.R.A.), AstraZeneca (to M.A.M.), and Natura (to M.A.M.) for funding this project. We thank the individuals or the individuals' families whose samples were used in this study.

## Author contributions

Conceptualization: D.M.-d.-S., M.K.O., L.O.L., and M.A.M. Methodology: T.D.S., F.M.-N., R.G.L., V.C.C., A.B.d.A.S., A.S.C.d.P., F.P.V., N.P.B.F., M.A.M.P., H.M.-S., T.L.K., G.P.R., G.S.P., I.M.P., and T.C.M.F.-C. Software: M.A.M.P. and M.B. Validation: T.D.S., F.M.-N., R.G.L., V.C.C., A.B.A.S., A.S.P., T.T.G., D.M.-d.-S., M.K.O., L.O.L., and M.A.M. Formal analysis: T.D.S., F.M.-N., R.G.L., V.C.C., A.B.A.S., A.S.P., F.P.V., T.T.G., M.A.M.P., B.J.S., D.M.-d.-S., M.K.O., L.O.L., and M.A.M. Investigation: T.D.S., F.M.-N., R.G.L., V.C.C., A.B.A.S., A.S.P., M.C.M., P.P.B., G.F.S., M.R.A., S.P.M., J.F., R.E.M., F.P.V., E.B., T.T.G., M.A.M.P., and T.T. Resources: N.P.B.F., C.M.K.O., R.B.M.J., P.H.C.d.A., S.S.B., R.M.M.V., D.M.d.M., A.T.F., E.A., T.M.C., F.Q.C., H.M.-S., M.B., M.A.R.V., A.S.F., P.M.M.M.-V., J.M.A.B., T.T., F.D.M.C., E.C., E.A.C., J.L.P.-M., D.M.-d.-S., M.K.O., L.O.L., and M.A.M. Data curation: T.D.S., F.M.-N., R.G.L., V.C.C., A.B.A.S., A.S.P., F.P.V., M.A.M.P., M.B., D.M.-d.-S., M.K.O., L.O.L., and M.A.M. Writing —original draft: T.D.S., F.M.-N., R.G.L., V.C.C., A.B.A.S., A.S.P., M.K.O., L.O.L., and M.A.M. Writing—review and editing: F.M.-N., R.E.M., H.M.S., M.A.R.V., P.M.M.M.-V., J.L.P.-M., and D.M.-d.-S. Visualization: T.D.S., F.M.-N., R.G.L., V.C.C., A.B.A.S., A.S.P., J.L.P.-M., D.M.-d.-S., M.K.O., L.O.L., and M.A.M. Supervision: A.T.F., E.A., F.Q.C., M.B., J.M.A.B., E.A.C., J.L.P.-M., D.M.-d.-S, M.K.O., L.O.L., and M.A.M. Project administration: D.M.-d.-S., M.K.O., L.O.L., and M.A.M. Funding acquisition: J.L.P.-M., D.M.-d.-S., M.K.O., L.O.L., and M.A.M.

## Competing interests

Marcelo A. Mori received funds from AstraZeneca and Natura for isolation and culture of adipose tissue mesenchymal stem cells for obesity- and aging-related research. The grants did not anticipate COVID-19 research. The remaining authors declare no competing interests.

## Ethics

This study was approved by local Research Ethics Committees and conducted by local researchers, and all of those who contributed significantly were included as authors. Engaging local researchers and sharing authorship among equal contributors have helped to stimulate a multidisciplinary environment, promote collaboration, and accelerate research locally in a period where the COVID-19 pandemics imposed many restrictions. Our research is particularly relevant locally since Brazil is one of the countries where obesity and severe COVID-19 have been most prevalent. Moreover, we studied the SARS-CoV-2 gamma variant of concern which has been initially identified in Brazil and led to hundreds of thousands of deaths and millions of people infected, mainly in South America. We have also cited local research relevant to our study whenever possible. Notably, most authors are from public universities in Brazil, which have been actively engaged in including the lower social strata among students and workers, following established long-term governmental policies.

## Additional information

Tatiana Dandolini Saccon[1,20], Felippe Mousovich-Neto [1,20], Raissa Guimarães Ludwig[1,20], Victor Corasolla Carregari[1,20], Ana Beatriz dos Anjos Souza[2,20], Amanda Stephane Cruz dos Passos [3,4,20], Matheus Cavalheiro Martini[5], Priscilla Paschoal Barbosa[5], Gabriela Fabiano de Souza[5], Stéfanie Primon Muraro [5], Julia Forato[5], Mariene Ribeiro Amorim[5], Rafael Elias Marques [6], Flavio Protasio Veras[3,4,7], Ester Barreto [3,4], Tiago Tomazini Gonçalves [4,8], Isadora Marques Paiva [3,4], Narayana P. B. Fazolini[1], Carolina Mie Kawagosi Onodera[9], Ronaldo Bragança Martins Junior[10], Paulo Henrique Cavalcanti de Araújo[2], Sabrina Setembre Batah[11], Rosa Maria Mendes Viana[10], Danilo Machado de Melo[10], Alexandre Todorovic Fabro[11], Eurico Arruda[10], Fernando Queiroz Cunha[3,4], Thiago Mattar Cunha [3,4], Marco Antônio M. Pretti [12], Bradley Joseph Smith[1], Henrique Marques-Souza [1], Thiago L. Knittel[1], Gabriel Palermo Ruiz[1], Gerson S. Profeta[1], Tereza Cristina Minto Fontes-Cal[1,13], Mariana Boroni[12,14], Marco Aurélio Ramirez Vinolo[5,13,14], Alessandro S. Farias [5,13,14], Pedro Manoel M. Moraes-Vieira [5,13,14], Joyce Maria Annichino Bizzacchi[9], Tambet Teesalu[15], Felipe David Mendonça Chaim [16], Everton Cazzo[16], Elinton Adami Chaim[16], José Luiz Proença-Módena[5,14], Daniel Martins-de-Souza [1,13,14,17,18] ✉, Mariana Kiomy Osako [2] ✉, Luiz Osório Leiria [3,4] ✉ & Marcelo A. Mori [1,13,14,19] ✉

[1]Department of Biochemistry and Tissue Biology, Institute of Biology, University of Campinas, Campinas, SP, Brazil. [2]Department of Cell and Molecular Biology and Pathogenic Bioagents, Ribeirão Preto Medical School, University of São Paulo, Ribeirão Preto, SP, Brazil. [3]Department of Pharmacology, Ribeirão Preto

Medical School, University of São Paulo, Ribeirão Preto, SP, Brazil. [4]Center for Research in Inflammatory Diseases, Ribeirão Preto, SP, Brazil. [5]Department of Genetics, Evolution, Microbiology and Immunology, Institute of Biology, University of Campinas, Campinas, SP, Brazil. [6]Brazilian Biosciences National Laboratory (LNBio), Brazilian Center for Research in Energy and Materials (CNPEM), Campinas, SP, Brazil. [7]Department of Biomolecular Sciences, School of Pharmaceutical Sciences of Ribeirão Preto, University of São Paulo, Ribeirão Preto, SP, Brazil. [8]Department of Pharmacology, Faculty of Medical Sciences, University of Campinas, Campinas, SP, Brazil. [9]Hematology-Hemotherapy Center, University of Campinas, Campinas, SP, Brazil. [10]Virology Research Center, Ribeirão Preto Medical School, University of São Paulo, Ribeirão Preto, SP, Brazil. [11]Department of Pathology and Legal Medicine, Ribeirão Preto Medical School, University of São Paulo, Ribeirão Preto, SP, Brazil. [12]Laboratory of Bioinformatics and Computational Biology, Division of Experimental and Transla-tional Research, Brazilian National Cancer Institute (INCA), Rio de Janeiro, Brazil. [13]Obesity and Comorbidities Research Center (OCRC), University of Campinas, Campinas, SP, Brazil. [14]Experimental Medicine Research Cluster (EMRC), University of Campinas, Campinas, SP, Brazil. [15]Laboratory of Precision and Nanomedicine, Institute of Biomedicine and Translational Medicine, University of Tartu, Tartu, Estonia. [16]Department of Surgery, Faculty of Medical Sciences, University of Campinas, Campinas, SP, Brazil. [17]D'Or Institute for Research and Education (IDOR), São Paulo, SP, Brazil. [18]Instituto Nacional de Biomarcadores em Neuropsiquiatria, Conselho Nacional de Desenvolvimento Científico e Tecnológico, São Paulo, SP, Brazil. [19]Instituto Nacional de Obe-sidade e Diabetes, Conselho Nacional de Desenvolvimento Científico e Tecnológico, Campinas, SP, Brazil. [20]These authors contributed equally: Tatiana Dandolini Saccon, Felippe Mousovich-Neto, Raissa Guimarães Ludwig, Victor Corasolla Carregari, Ana Beatriz dos Anjos Souza, Amanda Stephane Cruz dos Passos. ✉e-mail: dmsouza@unicamp.br; mko@fmrp.usp.br; luizleiria@usp.br; morima@unicamp.br

