## [Peer Review File · Nature Communications]

Title: SARS-CoV-2 infects adipose tissue in a fat depot- and viral lineage-dependent mannerREVIEWER COMMENTS

Reviewer #1 (Remarks to the Author):

The manuscript is interesting, relevant, and timely. The information is well presented in a logical and organized manner. My concerns involve statistical robustness of the proteomics data sets, which require further information and a re-analysis of the data with increased stringency.

Methods: Proteomics Sample Preparation

- Number of biological and technical replicates should be provided.
 - Provide information on the number of proteins and the date of download for the Homo sapiens uniprot FASTA file
 - It is standard within the field is to match a minimum of 2 peptides per protein for confident identification. The authors use a minimum of 1 peptide/protein. For robustness, I recommend reviewing the data sets and removing proteins identified with only 1 peptide match.
 - It is standard within the field to perform a multiple hypothesis correction using a false discovery rate (e.g., Benjamini-Hochberg test) on statistically significant calculations following an ANOVA or t-test. Typically, an FDR = 0.05 (or 0.01 for very stringent conditions) to account for a 5% error in significance with the ANOVA. This has not been done and is critical for a manuscript of this calibre. I recommend that the authors repeat the analysis following ANOVA with an FDR = 0.05 and re-assess the number of statistically significant proteins and the pathway/category enrichment analyses.
 - It is very good that the data has been deposited into PRIDE but the data is not searchable as it has not been publicly released. The reviewer login and password information needs to be provided.
- Ext. data Fig.4: suggest providing heatmaps in colours other than red/green to promote accessibility among readers and avoid issues with color blindness.
- Ext. data. Fig. 4: A-D – need y-axis. Suggest coloring significantly different proteins. Check if normalization (e.g., subtract median from each sample) is needed to provide a normal distribution with LFQ samples. Based on the skewing/heavy loading of protein spots to one side in A & B, this may be needed.

Reviewer #2 (Remarks to the Author):

Dandolini Saccon report that SARS-CoV-2 can infect human adipose tissue. Data is further backed up by infection of in vitro differentiated fat cells. Cell originating from visceral adipose tissue are a better source for infection than cells derived from subcutaneous adipose tissue, consistent with the epidemiological findings that puts patients with enhanced visceral adiposity at higher risk.

This is an intriguing set of observations that should be reported, despite the fact that other groups have highlighted adipose tissue as a site of infection. The authors also show that the virus productively

replicates in adipocytes, and highlight the fact that the stimulation of lipolysis enhances viral infection in adipose tissue.

The experiments are well performed, and the paper is very well written.

Specific points:

- Since the authors studied the effects of stimulation of lipolysis, it is unclear why they did not try to see whether a lipolysis inhibitor curbs viral spread.
- The authors do not comment on adiponectin, a protein that other groups have specifically focused on. What happened to adiponectin levels with adipose tissue in the different settings?
- A general weakness with all immunohistology is the absence of visualizing the adipocyte proper. Perilipin or caveolin stains should be used, or a fluorescent dye that partitions into the lipid droplet, such as BODIPY in non-fixed cells.
- I am not sure how meaningful the pathway analysis actually is, I think this part takes up an excessive amount of space.

Reviewer #3 (Remarks to the Author):

Overall, this is an interesting article addressing the role of fat as a source of SARS-CoV-2 replication.

- The detection of viral RNA in the adipose tissue of 54% of fatal cases – as this is simply reflective of viral RNA rather than active viral replication I am not sure what information this adds to the article
- The detection of spike would be more convincing but the IF are difficult to interpret – could co-staining be performed to better identify structures in the tissue – data shows one donor – how many is representative of?
- The statement (in the absence of data) that macrophages were infected with SARS-CoV-2 is difficult to assess as the general dogma in the literature is that macrophages can take up SARS-CoV-2 but as they do not express ACE2 it is not an actual infection
- It is surprising that viral RNA load did not correlate with BMI or weight – whilst this may be as the authors suggest a limitation of numbers are there other explanations they could offer
- In vitro studies are difficult because it is challenging to know in vivo what viral dose (if any) these cells are exposed to. Can the authors provide a rationale for the MOI used
- The IF Figures in Figure 2 (perhaps due to low quality) are not convincing that SARS-CoV-2 is able to replicate in Vis AD and Sub AD
- Figure 2 C and D confuse me – it appears there is a difference in viral replication between sub and vis AD if you look at RNA (C) but not if you look at infectious virus (D). Given that infectious virus is more relevant to the in vivo situation can the authors comment on this? This would seem to contradict their supposition that adipose tissue cells originated from visceral fat are intrinsically more permissive to SARS-CoV-2 infection than those from subcutaneous fat
- Expression of cell surface receptors is interesting but it is well known that RNA levels of these do not necessarily correlate with protein levels. These should be repeated by western blot or

immunofluorescence

- Given that proteomics was performed it would make more sense to validate the data on a protein level rather than on a RNA level
- The implication of this work is that in individuals with greater fat mass there would be increased SARS-CoV-2 replication – surely that data would be available as it would significantly strengthen the story

* Similarly, other respiratory viruses like IAV have increased severity in obese individuals - would you stipulate that a similar mechanism is at work here?

Response to the Reviewers

We thank the Reviewers for spending their time and borrowing their expertise to contribute to our manuscript. We appreciate the supportive comments and constructive suggestions, and hope to have addressed all of them in this revision. Please find point-by-point responses to your comments below where we indicate the changes made to the manuscript including new data added. To facilitate, we also attached to this submission a version of the revised manuscript with changes marked in blue. We believe the manuscript is much improved after the revision and hope the Reviewer agree that it is now ready for publication in the present format.

Reviewer #1 (Remarks to the Author):

The manuscript is interesting, relevant, and timely. The information is well presented in a logical and organized manner. My concerns involve statistical robustness of the proteomics data sets, which require further information and a re-analysis of the data with increased stringency.

R: We thank the Reviewer for finding the manuscript “interesting, relevant, and timely”, and for appreciating how the information is presented in it. We also thank the Reviewer for his/her fair review and constructive comments regarding the proteomics data, which prompted us to re-run and reanalyze the data to make it more statistically robust and even more informative. In brief, we optimized the electrospray ionization source and detector voltage and re-ran the samples in triplicate to increase statistical power. The optimization resulted in a much broader coverage of identified proteins and the higher number of replicates confirmed most of the statistical differences found in the previous analysis and revealed more enriched pathways. We have included the new data in the manuscript and made changes whenever relevant. Overall, the conclusions remained the same. Please find below the responses to each specific point raised by the Reviewer.

Methods: Proteomics Sample Preparation

- Number of biological and technical replicates should be provided.

R: We apologize for not making it clearer in the previous version of the manuscript. We have now included in the methods section and figure legends the number of biological and technical replicates used for the proteomic analyses.

- Provide information on the number of proteins and the date of download for the Homo sapiens uniprot FASTA file

R: We have now included the information in the Methods section.

- It is standard within the field is to match a minimum of 2 peptides per protein for confident identification. The authors use a minimum of 1 peptide/protein. For robustness, I recommend reviewing the data sets and removing proteins identified with only 1 peptide match.

R: We revised the data analysis according to the Reviewer’s suggestions. Since we re-ran the samples under optimized conditions and increased the number of technical replicates, we obtained a greater coverage of identified proteins even with the more stringent criteria. Information about the criteria used to identify proteins has been revised in the Methods section.

- It is standard within the field to perform a multiple hypothesis correction using a false discovery rate (e.g., Benjamini-Hochberg test) on statistically significant calculations following an ANOVA or t-test. Typically, an FDR = 0.05 (or 0.01 for very stringent conditions) to account for a 5% error in significance with the ANOVA. This has not been done and is critical for a manuscript of this calibre. I recommend that the authors repeat the analysis following ANOVA with an FDR = 0.05 and re-assess the number of statistically significant proteins and the pathway/category enrichment analyses.

R: In the revised version of the manuscript, we applied multiple hypothesis correction using the software Progenesis QI for Proteomics. The software allows for p-value correction using the characteristics of the distribution of calculated p-values, dynamically adjusting the p-values according to their spread. Information about the new statistical analysis was added to the manuscript. The method is explained in further detail by Nonlinear, the company behind Progenesis software (<https://www.nonlinear.com/support/progenesis/comet/faq/v2.0/pg-values.aspx>). Since we re-ran the samples under optimized conditions and increased the number of replicates, the number of statistically significant proteins and enriched pathways actually increased despite the more stringent statistical criteria. More importantly, the conclusions that were made based on the original data barely changed in comparison to the newly collected data; however, the information that can now be extracted from this analysis is much more statistically robust, and we thank the Reviewer for stimulating us to revise these analyses that helped us improve our manuscript. Figure 4 and Supplementary Figures 5-7 and Tables 2-3 containing the proteomics data were revised in the new version of manuscript.

- It is very good that the data has been deposited into PRIDE but the data is not searchable as it has not been publicly released. The reviewer login and password information needs to be provided.

R: For some reason, during the process of transferring the manuscript from one journal to another the login and password information to PRIDE was not made accessible to the Reviewers. Please find below the information for Reviewers' access of the data:

Project Name: SARS-CoV-2 infects adipose tissue in a fat depot- and viral lineage-dependent manner

Project accession: PXD034510

Project DOI: 10.6019/PXD034510

Username: reviewer_pxd034510@ebi.ac.uk

Password: NiUkwnCS

- Ext. data Fig.4: suggest providing heatmaps in colours other than red/green to promote accessibility among readers and avoid issues with color blindness.

R: Thank you for noticing this important aspect in this figure. We changed the heatmaps to red vs. blue in the new Supplementary Figure 6 as per the Reviewer's suggestion and hope that this will confer more accessibility among readers with color blindness.

- Ext. data. Fig. 4: A-D – need y-axis. Suggest coloring significantly different proteins. Check if normalization (e.g., subtract median from each sample) is needed to provide a normal distribution with LFQ samples. Based on the skewing/heavy loading of protein spots to one side in A & B, this may be needed.

R: Abundance data are normalized automatically to all proteins by Progenesis QI for Proteomics after sample alignment. We modified the volcano plots according to the Reviewer's suggestion and included them in new Supplementary Figure 5.

Reviewer #2 (Remarks to the Author):

Dandolini Saccon report that SARS-CoV-2 can infect human adipose tissue. Data is further backed up by infection of in vitro differentiated fat cells. Cell originating from visceral adipose tissue are a better source for infection than cells derived from subcutaneous adipose tissue, consistent with the epidemiological findings that puts patients with enhanced visceral adiposity at higher risk.

This is an intriguing set of observations that should be reported, despite the fact that other groups have highlighted adipose tissue as a site of infection. The authors also show that the virus productively replicates in adipocytes, and highlight the fact that the stimulation of lipolysis enhances viral infection in adipose tissue.

The experiments are well performed, and the paper is very well written.

R: We thank the Reviewer for finding our observations “intriguing” and worth reporting. We also appreciate his/her opinion that the experiments are “well performed” and the paper is “very well written”. We also thank him/her for the fair review and for the constructive suggestions that certainly helped us to improve the manuscript. Please find below the responses to each specific point raised by the Reviewer.

Specific points:

- Since the authors studied the effects of stimulation of lipolysis, it is unclear why they did not try to see whether a lipolysis inhibitor curbs viral spread.

R: Indeed, there is strong evidence provided by us and by other groups that lipolysis sustains viral replication in adipocytes. Here we demonstrate it by stimulating lipolysis with isoproterenol and finding more viruses in the culture medium. Consistent with our findings, Zickler et al. showed that blocking lipid breakdown using the lipase inhibitor tetrahydrolipstatin (Orlistat) reduces viral replication in adipocytes (doi: 10.1016/j.cmet.2021.12.002). Hence, we believe there is sufficient data in the manuscript and provided by others to confirm that lipolysis is both necessary and sufficient to stimulate viral replication in adipocytes. Confident that our data complemented observations from other groups that used a lipase inhibitor to demonstrate the involvement of lipid breakdown in SARS-CoV-2 replication in adipocytes, we thought repeating this experiment exactly how it was reported was not necessary. We hope the Reviewer agrees and is convinced, after all, with the role of lipolysis in the viral spread in adipocytes.

- The authors do not comment on adiponectin, a protein that other groups have specifically focused on. What happened to adiponectin levels with adipose tissue in the different settings?

R: The Reviewer calls attention to yet another relevant observation from the literature, considering that other groups did observe decreased adiponectin levels in patients with COVID-19 and hamsters infected with SARS-CoV-2. As per the Reviewer’s suggestion, we looked at adiponectin levels in our proteomics datasets. We identified adiponectin in the datasets of subcutaneous adipose tissue cells and found that its protein levels were decreased by approximately 15% upon SARS-CoV-2 infection (B lineage), although at

the borderline of statistical significance (q value = 0.06). This somewhat confirms the literature, although the effect size is small and barely significant in our model. We included reference to the data in Supplementary Table 2 and a brief discussion about the topic in the revised version of the manuscript.

- A general weakness with all immunohistology is the absence of visualizing the adipocyte proper. Perilipin or caveolin stains should be used, or a fluorescent dye that partitions into the lipid droplet, such as BODIPY in non-fixed cells.

R: Based on the Reviewer's suggestion, we have counterstained perilipin 1 and 2 and provided bright-field images of the postmortem adipose tissue samples for better visualization of adipocytes. In addition, we used LipidTOX to stain lipid droplets in the differentiated adipocytes *in vitro*. We thank the Reviewer for making this comment since it allowed us to revise our immunofluorescence analyses and make them even more convincing that adipocytes are infected by SARS-CoV-2. The new data have now been included in new Figures 1A, 2B, 3A and Supplementary Figures 1 and 4.

- I am not sure how meaningful the pathway analysis actually is, I think this part takes up an excessive amount of space.

R: We agree with the Reviewer that the description of the enriched pathways could be shortened and we did that in the revised version of the manuscript.

Reviewer #3 (Remarks to the Author):

Overall, this is an interesting article addressing the role of fat as a source of SARS-CoV-2 replication.

R: We thank the Reviewer for finding our article of interest and for pointing out aspects of the work that could be improved to make it even more convincing. We also thank him/her for the fair review and constructive suggestions that certainly helped us to improve the manuscript. Please find below the responses to each specific point raised by the Reviewer.

- The detection of viral RNA in the adipose tissue of 54% of fatal cases – as this is simply reflective of viral RNA rather than active viral replication I am not sure what information this adds to the article

R: We acknowledge the limitations of using viral RNA to detect viable viruses in biological samples, but our decision to use this assay to screen the presence of SARS-CoV-2 in adipose tissue was based on the fact that this is a highly sensitive approach normally used to provide a first line of investigation regarding the presence or not of the virus in a given sample. Moreover, we leaned on previous publications, especially the one by Yao et al. 2021, published in Cell Research (doi: 10.1038/s41422-021-00523-8), in which the authors established SARS-CoV-2 organotropism by comparing the presence of viral RNA in multiple organs obtained during the autopsy of individuals who died of COVID-19. Using this approach, they concluded that 46% of individuals deceased from COVID-19 had systemic distribution of viral RNA, whereas the extrapulmonary tissue where SARS-CoV-2 RNA was more frequently identified was the aorta (36%), followed by the small intestine (31%). Importantly, as far as our work is concerned, they did not investigate adipose tissue in their study. We found that 49% of individuals who died of COVID-19 had viral RNA detected in adipose tissue samples, indicating that adipose tissue is a major extrapulmonary site for the virus if compared to the 19 other non-pulmonary tissues evaluated by Yao et al. Although differences in assay sensitivity could account for these differences, our observations were confirmed by Zickler et al. (2022) using an independent cohort of individuals (doi: 10.1016/j.cmet.2021.12.002). Similar to the number we obtained, they detected the presence of SARS-CoV-2 RNA in adipose tissue samples of 50% of subjects who died of COVID-19. Furthermore, in a more recent paper, Basolo et al. detected SARS-CoV-2 RNA in 13/23 (56%) of subcutaneous abdominal adipose tissue specimens of subjects deceased from COVID-19 (doi: 10.1007/s40618-022-01742-5). Importantly, they went on to show that 12/12 COVID-19 cases had the virus nucleocapsid antigen detected in adipocytes. Hence, we believe that the information about the proportion of deceased individuals harboring SARS-CoV-2 RNA in adipose tissue is in fact relevant to strengthen the notion supported by the literature that adipose tissue is a major extrapulmonary site where SARS-CoV-2 can be found. Complementary to it, we have immunofluorescence data showing that the spike protein can also be detected in postmortem adipose tissue samples and in vitro data showing that SARS-CoV-2 can indeed infect and replicate in adipose tissue cells. Altogether, we believe we have sufficient data to demonstrate that SARS-CoV-2 can reach adipose tissue, infect it and replicate in it. Still, considering the Reviewer's comment, we decided to tone down the conclusions that could be drawn by SARS-CoV-2 RNA detection and revised the manuscript accordingly.

- The detection of spike would be more convincing but the IF are difficult to interpret – could co-staining be performed to better identify structures in the tissue – data shows one donor – how many is representative of?

R: Based on the Reviewer's suggestion, we have counterstained perilipin 1 and 2 and provided bright-field images of the postmortem adipose tissue samples for better visualization of the tissue structures, particularly the adipocytes. This led to the new Figures 1A and Supplementary Figure 1. In our opinion, the images are much more convincing and clearly demonstrate the widespread distribution of spike in adipose tissue, where adipocytes appear to be among the infected cells. In addition, we used LipidTOX to stain lipid droplets in the differentiated adipocytes in vitro and confirmed that adipocytes are indeed infected by SARS-CoV-2. This new data has now been included in new Figures 2B and Supplementary Figure 4. Altogether, we think there is strong evidence to support the notion that SARS-CoV-2 viral particles can be detected in adipose tissue cells, particularly in adipocytes, and the virus can infect and replicate in these cells. Regarding the number of donors, we indicated in the figure legend that the images represent immunofluorescence images of 6-8 frames of 2 individuals chosen based on the quality of the histological slides. In both individuals, we found the presence of SARS-CoV-2 spike in adipose tissue.

- The statement (in the absence of data) that macrophages were infected with SARS-CoV-2 is difficult to assess as the general dogma in the literature is that macrophages can take up SARS-CoV-2 but as they do not express ACE2 it is not an actual infection

R: Since we did not focus on macrophages in this article, and considering that, as commented by the Reviewer, it remains unclear whether macrophages are actually infected by or simply take up SARS-CoV-2, we decided to remove from the current version of the manuscript the sentence where we mentioned our observation that macrophages could be infected by SARS-CoV-2. We also avoided the word "infection" when discussing how macrophages can contribute to adipose tissue inflammation when bearing SARS-CoV-2. We believe these changes do not compromise the manuscript, which is focused on adipocyte lineages and their intrinsic differences between fat depots.

- It is surprising that viral RNA load did not correlate with BMI or weight – whilst this may be as the authors suggest a limitation of numbers are there other explanations they could offer

R: As the Reviewer, at first, we were also surprised by the fact that viral RNA load did not correlate with BMI. Although the sample size could be an explanation, what our data is exactly telling us is that the potential of SARS-CoV-2 to infect subcutaneous adipose tissue from the thoracic region occurs independently of BMI. This is because we measured the relative abundance between SARS-CoV-2 RNA and host tissue RNA. Hence, if considering the amount of host RNA proportional to the number of cells, one could suggest that adipose tissue hyperplasia in obese individuals could still provide a larger reservoir for the virus. Moreover, as mentioned by the Reviewer, viral RNA does not necessarily reflect active viral replication. Still, obese individuals could have signals that favor viral replication or persistence in adipose tissue, but whether this is the case will have to be explored in a follow-up study. Finally, BMI does not accurately reflect adipose tissue mass distribution and our study demonstrates that

cells from different fat depots enable SARS-CoV-2 infection differently. Indeed, based on our study, one could predict that viral load could be more associated with visceral adiposity rather than subcutaneous adiposity. However, we could only access fat samples from the thoracic region to assess the viral load, hence, our conclusions are limited to this tissue. All these limitations were pointed out in the revised version of the manuscript.

- In vitro studies are difficult because it is challenging to know in vivo what viral dose (if any) these cells are exposed to. Can the authors provide a rationale for the MOI used

R: We agree with the Reviewer about the limitations of in vitro studies to reflect in vivo conditions. However, the use of a reductionist approach may be useful to address specific questions. In this case, after we observed that adipose tissue can be infected *in vivo*, we decided to use the *in vitro* system to compare the intrinsic capacity of cells isolated from different fat depots of the same individuals to be infected by SARS-CoV-2. For that reason we chose a MOI of 1.0, so that virus availability did not represent a limiting factor and we could properly assess the intrinsic capacity of adipose tissue cells to be infected. This rationale was included in the revised version of the manuscript. These experiments allowed us to conclude that visceral fat cells are more susceptible to SARS-CoV-2 infection. The only occasion when we used a lower MOI was when performing immunofluorescence for this revision. In this case, we wanted the infection to occur only in a limited number of cells so that we could gain contrast and resolution in our images. In conclusion, despite the normal limitations of using *in vitro* systems, our experiments provided strong evidence that adipose tissue cells can be infected by SARS-CoV-2 and the virus can replicate in these cells. In addition, we can conclude that visceral fat cells are more susceptible to SARS-CoV-2 infection when compared to subcutaneous fat cells.

- The IF Figures in Figure 2 (perhaps due to low quality) are not convincing that SARS-CoV-2 is able to replicate in Vis AD and Sub AD

R: As per the Reviewer's suggestion, we repeated the immunofluorescence in Vis AD and Sub AD cells including LipidTOX staining and the bright field. We believe the images are much improved and clearly evidence co-localization of SARS-CoV-2 spike and dsRNA in adipocytes, supportive of viral replication in these cells (see new Figure 2B and Supplementary Figure 4). The capacity of SARS-CoV-2 to replicate in adipose tissue cells is further supported by the increase in viral load over time and the identification of infectious particles in the medium.

- Figure 2 C and D confuse me – it appears there is a difference in viral replication between sub and vis AD if you look at RNA (C) but not if you look at infectious virus (D). Given that infectious virus is more relevant to the in vivo situation can the authors comment on this? This would seem to contradict their supposition that adipose tissue cells originated from visceral fat are intrinsically more permissive to SARS-CoV-2 infection than those from subcutaneous fat

R: We apologize if the figures were not sufficiently clear. In fact, when considering the ancient SARS-CoV-2 B strain, the amount of infectious particles produced by Vis AD cells is approximately 770-fold higher than that of Sub AD cells (Figure 2D, 1st bar compared to 3rd bar, please note that the values in the graph are in Log10). This

difference is within a similar range (i.e., 240-fold higher in Vis AD compared to Sub AD) when looking at viral load (Figure 2C, 3rd bar vs. 7th bar, also in Log10). Hence, the results are in agreement with evidence that visceral fat cells are more susceptible to SARS-CoV-2 infection. Perhaps what confused the Reviewer was the results obtained with the P.1 variant. In this case, both P.1 RNA and the quantity of infectious particles were lower when compared to the values found to the B strain in Vis AD cells (Figure 2C, 7th vs. 8th bars; Figure 2D, 3rd vs. 4th bars). Since this attenuation was not found in Sub AD, it reduced the differences found between Vis AD and Sub AD cells in regards to P.1 viral load and infectious particles (Figure 2C, 4th vs. 8th bars; Figure 2D, 2nd vs. 4th bars). Still, the viral load and the amount of infectious particles of P.1 remained a little over one order of magnitude higher in Vis AD when compared to Sub AD, although for the infectious particles the difference did not reach statistical significance (p value = 0.0935) due to interindividual variability. We included this p value in the revised version of the manuscript to call attention to this trend. Regardless, we believe that these marginal differences when looking at the P.1 strain do not contradict our observations, but rather strengthen our conclusion of an attenuated infection by the P.1 variant in Vis AD cells. These results also demonstrate that regardless of the infectious capacity of the SARS-CoV-2 strain, visceral fat cells remain more susceptible to infection when compared to subcutaneous fat cells.

- Expression of cell surface receptors is interesting but it is well known that RNA levels of these do not necessarily correlate with protein levels. These should be repeated by western blot or immunofluorescence

R: The Reviewer is right. Although mRNA data is indicative, protein data is more adequate to assess the actual levels of cell surface receptors. Based on the Reviewer's suggestion, we performed immunofluorescence to qualitatively and topographically measure ACE2 in our cell culture models (new Figure 3A). We found that ACE2 is indeed expressed by adipose tissue cells, including in lipid droplet-containing cells. To quantify ACE2 levels, we performed western blotting and compared Sub AD and Vis AD cells (new Figure 3B). In agreement with the mRNA data, Vis AD cells expressed more ACE2 when compared to Sub AD cells. Altogether, these results confirm that ACE2 can serve as a gateway to SARS-CoV-2 in adipose tissue cells and that visceral fat cells express more ACE2, which is consistent with the higher susceptibility of these cells to SARS-CoV-2 infection.

- Given that proteomics was performed it would make more sense to validate the data on a protein level rather than on a RNA level

R: We tried to find ACE2 in our proteomics data but the method was not sensitive enough to detect it. For that reason, we used western blotting as explained before.

- The implication of this work is that in individuals with greater fat mass there would be increased SARS-CoV-2 replication – surely that data would be available as it would significantly strengthen the story

R: We agree with the Reviewer that if data like that happened to be available, it would strengthen our story. However, to be truly supportive of our story, this data should necessarily include viral load and other parameters of viral replication in visceral fat. Moreover, visceral fat mass should be measured, not simply BMI. Finally, the *n* should

be large enough to allow robust statistical associations. A study like that is challenging, because it has to involve a large number of postmortem samples or samples from individuals undergoing abdominal surgery while infected with SARS-CoV-2. While the latter is uncommon, the former could be possible. However, parameters to assess visceral fat mass would be less likely to be available for these individuals. Although not impossible, a study like that requires time and dedicated resources, and in our opinion, should be conducted independently as a follow up. In fact, our data, if published, could encourage groups to embark in such an enterprise. Finally, we would like to make the point that although one important implication of our work is the association between visceral fat mass and the increased potential for SARS-CoV-2 infection and replication, as stated by the Reviewer, another equally relevant clinical implication of our study is the fact that SARS-CoV-2 infection elicits a much more potent inflammatory response in visceral fat when compared to subcutaneous fat, which, in consonance with the contribution of visceral fat to systemic inflammation during metabolic diseases, could explain why visceral fat mass (rather than subcutaneous fat or overall adiposity) is a better predictor of COVID-19 severity risk. For such an observation there are plenty of supporting studies available in the literature, which together with our findings, strengthen the relevance of visceral fat infection in the pathophysiology of COVID-19.

* Similarly, other respiratory viruses like IAV have increased severity in obese individuals - would you stipulate that a similar mechanism is at work here?

R: We thank the Reviewer for bringing up this point and allowing us to speculate about other viruses that may use a similar mechanism to influence disease pathogenesis in obese individuals. Indeed, it would be interesting to see future studies comparing the behavior of adipose tissue SARS-CoV-2 infection to other viruses that have been shown to infect fat cells (e.g., influenza A virus, human immunodeficiency virus, adenovirus), as well as to investigate whether the way obesity influences the pathogenesis of infectious diseases other than COVID-19 (e.g., influenza) is also linked to the abundance of visceral vs. subcutaneous fat or the differential susceptibility to infection these fat depots may exhibit. Stimulated by the Reviewer, we included this discussion in the revised version of the manuscript.

REVIEWERS' COMMENTS

Reviewer #1 (Remarks to the Author):

I am impressed and pleased with the author's responses to my comments and greatly appreciate the effort put in to increase the number of samples measured through additional technical replicates. In addition, the attention to re-processing the data with increased statistical robustness is admirable and supports the thoroughness and integrity of the authors and the study.

Reviewer #2 (Remarks to the Author):

The authors have appropriately answered the concerns raised in the original review.

Reviewer #3 (Remarks to the Author):

The authors have done an excellent job addressing all the reviewers concerns and I think the paper as is makes a substantial contribution to the litterature and is appropriate for publication.

Response to the Reviewers

Reviewer's comments:

Reviewer #1 (Remarks to the Author):

I am impressed and pleased with the author's responses to my comments and greatly appreciate the effort put in to increase the number of samples measured through additional technical replicates. In addition, the attention to re-processing the data with increased statistical robustness is admirable and supports the thoroughness and integrity of the authors and the study.

Reviewer #2 (Remarks to the Author):

The authors have appropriately answered the concerns raised in the original review.

Reviewer #3 (Remarks to the Author):

The authors have done an excellent job addressing all the reviewers concerns and I think the paper as is makes a substaintial contribution to the litterature and is appropriate for publication.

We thank the Reviewers for their constructive feedback and for acknowledging our effort to revise the manuscript. We are happy to hear that all three Reviewers are now satisfied with the current version of the manuscript and recommend publication.